# An interactive deep learning-based approach reveals mitochondrial cristae topologies

**Shogo Suga**[1], **Koki Nakamura**[1], **Yu Nakanishi**[1], **Bruno M. Humbel**[2,3], **Hiroki Kawai**[1]*, **Yusuke Hirabayashi**[1,4]*

1 Department of Chemistry and Biotechnology, School of Engineering, The University of Tokyo, Tokyo, Japan, 2 Imaging Section, Okinawa Institute of Science and Technology (OIST), Okinawa, Japan, 3 Department of Cell Biology and Neuroscience, Juntendo University Graduate School of Medicine, Tokyo, Japan, 4 Department of Bioengineering, School of Engineering, The University of Tokyo, Tokyo, Japan

☯ These authors contributed equally to this work.
* hkawai@g.ecc.u-tokyo.ac.jp (HK); hirabayashi@chembio.t.u-tokyo.ac.jp (YH)

**Data Availability Statement:** All raw data for quantification summary of the electron microscopy images are within the paper and its Supporting Information files. All the original electron

## Abstract

The convolution of membranes called cristae is a critical structural and functional feature of mitochondria. Crista structure is highly diverse between different cell types, reflecting their role in metabolic adaptation. However, their precise three-dimensional (3D) arrangement requires volumetric analysis of serial electron microscopy and has therefore been limiting for unbiased quantitative assessment. Here, we developed a novel, publicly available, deep learning (DL)-based image analysis platform called **P**ython-based **h**uman-**i**n-the-**lo**op **w**orkflow (PHILOW) implemented with a human-in-the-loop (HITL) algorithm. Analysis of dense, large, and isotropic volumes of focused ion beam-scanning electron microscopy (FIB-SEM) using PHILOW reveals the complex 3D nanostructure of both inner and outer mitochondrial membranes and provides deep, quantitative, structural features of cristae in a large number of individual mitochondria. This nanometer-scale analysis in micrometer-scale cellular contexts uncovers fundamental parameters of cristae, such as total surface area, orientation, tubular/lamellar cristae ratio, and crista junction density in individual mitochondria. Unbiased clustering analysis of our structural data unraveled a new function for the dynamin-related GTPase Optic Atrophy 1 (OPA1) in regulating the balance between lamellar versus tubular cristae subdomains.

## Introduction

Optimizing the metabolic capacity of mitochondria requires convolution of the inner mitochondrial membrane (IMM) into cristae [1]. Cristae are specialized IMM infoldings where the entire set of 5 macromolecular complexes underlying the electron transport chain (ETC) and oxidative phosphorylation (OxPhos) are localized [2,3]. Since the narrow compartment of cristae represents a significant bottleneck for the diffusion of ions and metabolites, proper control of crista structure is important for metabolic homeostasis [4–7]. Despite their critical importance, quantitative and unbiased measurements of the structural features of IMM and cristae still represent a major roadblock for the study of mitochondrial biology.

microscopy images are available from a deposit site below. https://www.ebi.ac.uk/empiar/EMPIAR-11449/ All original code has been deposited at https://github.com/neurobiology-ut/PHILOW and is publicly available as of the date of publication. Plasmids encoding generated in this study have been deposited to Addgene (plasmid numbers 206316 and 206319).

**Funding:** This work was supported by following financial sources. The Japan Society for the Promotion of Science (https://www.jsps.go.jp/english/index.html) KAKENHI under Grant Number 20H04898 (Y.H.), Japan Agency for Medical Research and Development (https://www.amed.go.jp/index.html) under Grant number JP19dm0207082 (Y.H.), Basis for Supporting Innovative Drug Discovery and Life Science Research (BINDS) from AMED under grant numbers 19am0101116j0003 (B.M.H.) and 20am0101116j0004 (B.M.H.), The Japan Society for the Promotion of Science KAKENHI under Grant Number 22J23099 (K.N.), 22J23115 (S.S.), 21K19253 (Y.H.), 20K22622 (H.K.), SECOM Science and Technology Foundation Research grant (https://www.secomzaidan.jp/) (Y.H.), and the Uehara memorial foundation research grant (https://www.ueharazaidan.or.jp/) (Y.H.) and Chan Zuckerberg initiative napari Ecosystem Grants (https://chanzuckerberg.com/science/programs-resources/imaging/napari/) (H.K.). The funders had no role in study design, data collection and analysis, decision to publish, or preparation of the manuscript.

**Competing interests:** I have read the journal's policy and the authors of this manuscript have the following competing interests: HK is an employee of LPIXEL Inc.

**Abbreviations:** 3D, three-dimensional; BCE-dice-loss, binary cross-entropy dice coefficient loss; BSE, backscattered electron; CJ, crista junction; DL, deep-learning; DMEM, Dulbecco's Modified Eagle Medium; ET, electron tomography; ETC, electron transport chain; FIB-SEM, focused ion beam-scanning electron microscopy; GT, ground truth; GUI, graphical user interface; HITL, human-in-the-loop; IBM, inner boundary membrane; IMM, inner mitochondrial membrane; IoU, intersection over union; MCI, mitochondrial complexity index; OMM, outer mitochondrial membrane; OPA1, optic atrophy 1; OxPhos, oxidative phosphorylation; PC, principal component; PCA, principal component analysis; PHILOW, Python-based human-in-the-loop workflow; SIFT, scale-invariant feature transform; sSEM, serial scanning EM; TAP, three-axes prediction; TEM, transmission electron microscopy; TLD, through-the-lens detector.

Transmission electron microscopy (TEM), especially electron tomography (ET), has played pivotal roles in describing the ultrastructural features of IMM organization because of its unparalleled resolution. It revealed that cristae can adopt either flat structure (lamellar cristae) or tube (tubular cristae) [8,9]. For example, mitochondria in cardiomyocytes have densely packed lamellar cristae, while neuronal mitochondria are largely composed of tubular cristae. However, the nanometer-size volume of ET was not sufficient for resolving the entire crista structure in micrometer-scale individual mitochondria. In order to fill this gap in scale between crista structure and the organelle itself, techniques combining serial sectioning and scanning EM (serial scanning EM, sSEM; SEM tomography) have been developed. In particular, focused ion beam-scanning electron microscopy (FIB-SEM) lends itself well to the investigation of ultrastructural features in 3D because of its isotropic X-Y-Z resolution in the low nanometer range [10–14]. In addition, FIB-SEM covers volumes with tens of micrometers in scale.

With the increased throughput of image acquisition in FIB-SEM and related 3D EM approaches comes an unsolved challenge with limited throughput and accuracy in image segmentation. Because EM visualizes virtually all membrane structures in grayscale, segmenting structures of interest is an indispensable and highly challenging process. In most image analyses, defining thresholds for differentiating an object of interest over background or other structures is an essential step. Although computer-based image analyses have expanded parameters for defining objects, it has been challenging to determine proper thresholding values for extracting an object of interest. Even with traditional machine learning, it remains challenging to define thresholds for segmenting objects from images with diverse textures and especially with complex 3D structures. Recent advances in the application of deep learning (DL), which does not require thresholding for image classification, object detection, and pixel segmentation, are expected to transform image analysis in biological studies with versatility and efficiency [15]. However, while image processing speed increased, laborious manual processing, such as generating training data, proofreading prediction results, and converting files for processing images across multiple software programs, hampers application of DL for solving actual biological questions.

To provide a general solution to this problem, we developed a new integrated platform called **P**ython-based **h**uman-**i**n-the-**lo**op **w**orkflow (PHILOW), equipped with active learning for efficient iterative training data generation, and a new 3D structure prediction algorithm using 2D training datasets. Implementation of this platform drastically reduced the amount of human labor required for segmentation while increasing the precision of segmentation. This allowed high-throughput cristae analysis at nanometer resolution in micrometer scale. We successfully reconstructed a comprehensive structure of mitochondria and cristae from 135 control and 324 OPA1-deficient mitochondria. This unprecedented nanoscale ultrastructural analysis in a cellular context determined the total surface area, orientation, and spatial arrangements of individual mitochondria. It also revealed a previously unappreciated abundance of tubular crista structures in a mouse fibroblast cell line. Using this comprehensive structural analysis, we also revealed novel roles for OPA1, a protein best characterized for its role in IMM fusion, in the regulation of crista ultrastructural feature.

## Results

### Highly accurate DL-based segmentation requires limited proofreading

Since the analysis of cellular ultrastructure necessitates highly accurate segmentation, DL-based segmentation requires labor-intensive manual proofreading [16]. We first examined the relationship between proofreading time and accuracy (F1 score; the harmonic mean of

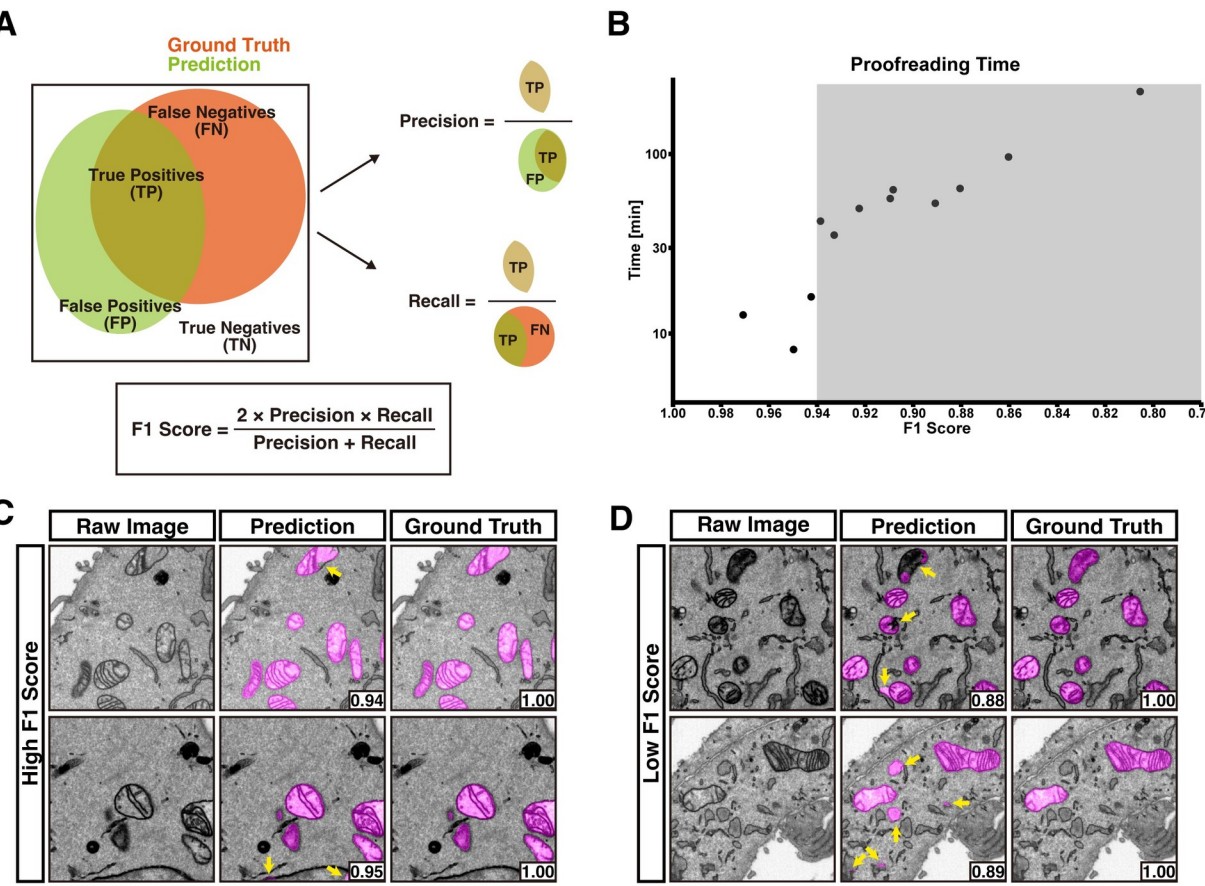

**Fig 1. Relationship between F1 score and proofreading time. (A)** A diagram explaining the F1 score. **(B)** Times required for proofreading $100 \times 512 \times 512$ voxel mitochondrial predictions with various F1 scores until the scores exceed 0.96 were measured. Note that the y-axis is presented in log scale. When the F1 score is less than 0.94 (highlighted with gray), the proofreading time jumps up. Source data can be found in **S1 Data**. **(C, D)** Representative images from stacks with high **(C)** and low **(D)** F1 score. Yellow arrows indicate areas requiring manual correction. Images with high F1 score have less and simpler mis-prediction than images with low F1 score. The F1 score of each crop is shown at the right bottom. The raw EM data are deposited in the EMPIAR (EMPIAR-11449). EMPIAR, Electron Microscopy Public Image Archive; EM, electron microscopy.

precision and recall, **Fig 1A**) of mitochondrial predictions. Measurement of proofreading times required for DL-based mitochondrial segmentations with various F1 scores revealed that segmentations with F1 scores >0.94 required only less proofreading time (**Fig 1B and 1C and S1 Data**). On the other hand, the proofreading times for segmentations with F1 scores <0.94 significantly increased (**Fig 1B and 1D and S1 Data**). Therefore, in this study, we aimed to develop a method to create a model whose prediction achieves a F1 score greater than 0.94.

## Three-axes prediction (TAP) reduces section-to-section inconsistency of inference

When mitochondrial structures were predicted by a conventional 2D UNet++ model [17], the prediction yielded an F1 score mostly under 0.94. To improve the F1 score, we first aimed to reduce section-to-section inconsistency, which is ascribed to inaccurate predictions at the boundaries of mitochondria (**S1 Fig**). We reasoned that inference from a side view of the image volume would decrease the inconsistency and thus developed a three-axes prediction (TAP) method (**Fig 2A**). With this method, a prediction model trained with a xy-plane dataset

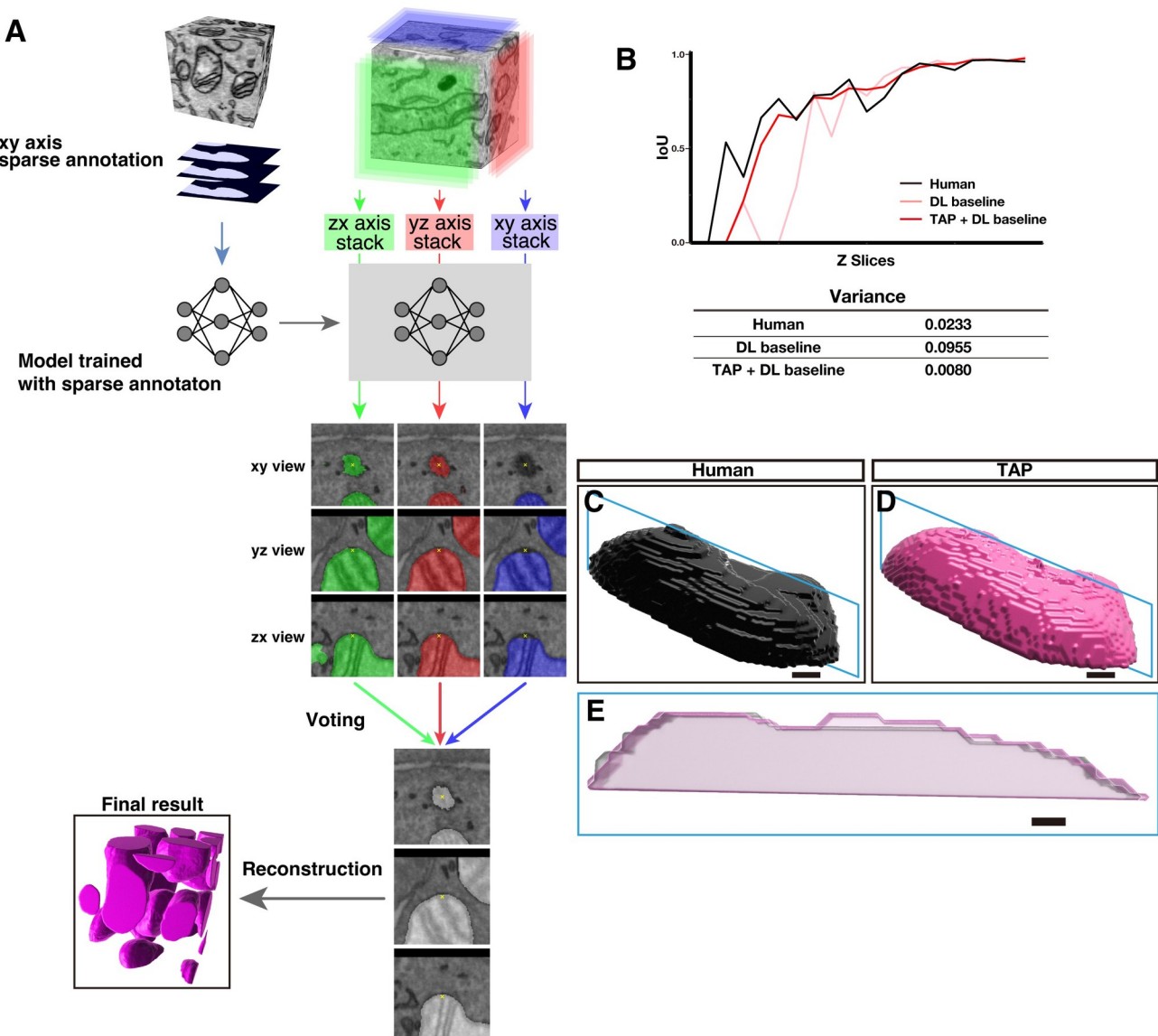

**Fig 2. TAP method enables a precise segmentation at the periphery of mitochondria. (A)** A diagram explaining the TAP method. A model trained with sparse annotation on xy-plane images was applied for the virtual yz (red)- and zx (green)-plane images as well as xy (blue)-plane images. Therefore, each voxel has 3 prediction results from 3 axes. An example of 3 predictions (green, red, and blue) visualized from 3 axes are shown in the middle. A majority vote was taken from the 3 predictions, and the result was adopted as the true value. **(B)** The IoU overlap with the next slice was measured for mitochondria segmentation of each slice from manual segmentation (black) or in the prediction either using the TAP + DL baseline (red) or only DL baseline (pink). The variances of the IoU are shown in the bottom table. Source data can be found in **S2 Data. (C, D)** 3D reconstructions of a part of mitochondria segmented by a human (C) and TAP method (D). **(E)** An overlay of cross-sectional views at the zx-plane (blue plane in C, D). Note that the human annotations missed the right-top area and were anomalous at the left. Scale bar, 100 nm. The raw EM data are deposited in the EMPIAR (EMPIAR-11449). EMPIAR, Electron Microscopy Public Image Archive; EM, electron microscopy; TAP, three-axes prediction; 3D, three-dimensional; IoU, intersection over union; DL, deep learning.

is applied to virtual yz and zx-planes. Especially, when an image volume consists of isotropic voxels, a model generated from xy-planes is directly applied to additional 2 planes and intermediate predictions from 3 axes are produced. By majority vote of the intermediate predictions, a final prediction at each voxel is determined. When TAP was applied to segment mitochondria from a $10 \times 10 \times 10$ nm isotropic volume ($309 \times 732 \times 1,554$ voxels, 10 out of

309 slices were used for training and all 309 slices were applied for test), strikingly, the variance of intersection over union (IoU) between neighboring xy-slices of inferences was 12 times lower compared to that without TAP (0.0080 and 0.0955 with or without TAP, respectively) and 3 times lower than manual segmentation (0.0233) (**Fig 2B** and **S2 Data**). This showed that boundaries determined by the TAP method are more consistent along serial images and even superseded the human expert (**Fig 2C–2E**). As a result of introducing TAP, the F1 score was increased to the level we aimed for (0.96 with TAP, 0.94 without TAP).

## Efficient choice of training areas by human-in-the-loop iterative method

Among current limitations in generating high-performance DL models is a difficulty in obtaining training datasets covering highly diverse features that represent objects of interest. Although drawing support tools have accelerated manual segmentation [18], choice of areas for training datasets has been random, which leads to redundant annotations of similar structures and ignorance of rare structures. To collect training data covering comprehensive features efficiently, we developed a DL scheme by employing human-in-the-loop (HITL) and pixel-level active learning (**Fig 3A**). To test the HITL scheme, we segmented mitochondria from $248 \times 1,147 \times 960$ voxels FIB-SEM images of NIH3T3 cells at $10 \times 10 \times 10$ nm per voxel. After the first round of learning with 3 manually segmented training datasets, the DL model successfully segmented structures with major mitochondrial features but failed to segment rare features (first prediction, **Fig 3B**). Subsequently, 3 sections with suboptimal prediction accuracy were identified and their predictions were manually corrected. The corrected sections were then incorporated as additional training data to the initial 3 training data sections for retraining the model (second prediction, **Fig 3B**). Another round of iteration with 4 more (a total of 10 out of 248) areas of the HITL-learning generated a DL model covering mitochondrial features highly comprehensively. The F1 score compared to the ground truth (GT) was calculated for the predicted results for entire slices, including the slices used as the training data, and reached 0.97 with the constant increase along three iterative cycles (0.906 after the first cycle and 0.924 after the second cycle), whereas a model generated from random 10 training areas exhibited a much lower F1 score (0.909). Among other advantages, this high F1 score reflects the appropriate separations of 2 mitochondria, which are often too closely apposed to be separated without specialized post-processing in previously reported automated mitochondrial segmentation or low cycles of HITL (Slice #4 in **S1B Fig**). These results suggest that the HITL-assisted choice of image areas for generating training datasets significantly increases the performance of models without increasing the volume of training data.

## Improvement of mitochondrial segmentation by the HITL approach on PHILOW

A roadblock for implementing the HITL-iterative scheme was a complicated set of file format conversions, export and import cycles of data across different applications, and accompanying file management. Thus, to circumvent this, we developed an open-source integrated analysis platform called PHILOW, equipped with a seamless graphical user interface (GUI) environment for annotation assistance, visualization, data management, model training, inference, and manual correction functions (**S2 Fig**). Also, by cloud-based model training and prediction, PHILOW enables an easy introduction of DL-based analyses without purchasing GPUs and associated software. PHILOW is available at github (https://github.com/neurobiology-ut/PHILOW) and seamlessly incorporated into napari as a plugin.

To evaluate the effect of seamless HITL implementation using PHILOW, we measured the total work time required for a complete 3D reconstruction of mitochondrial structures from

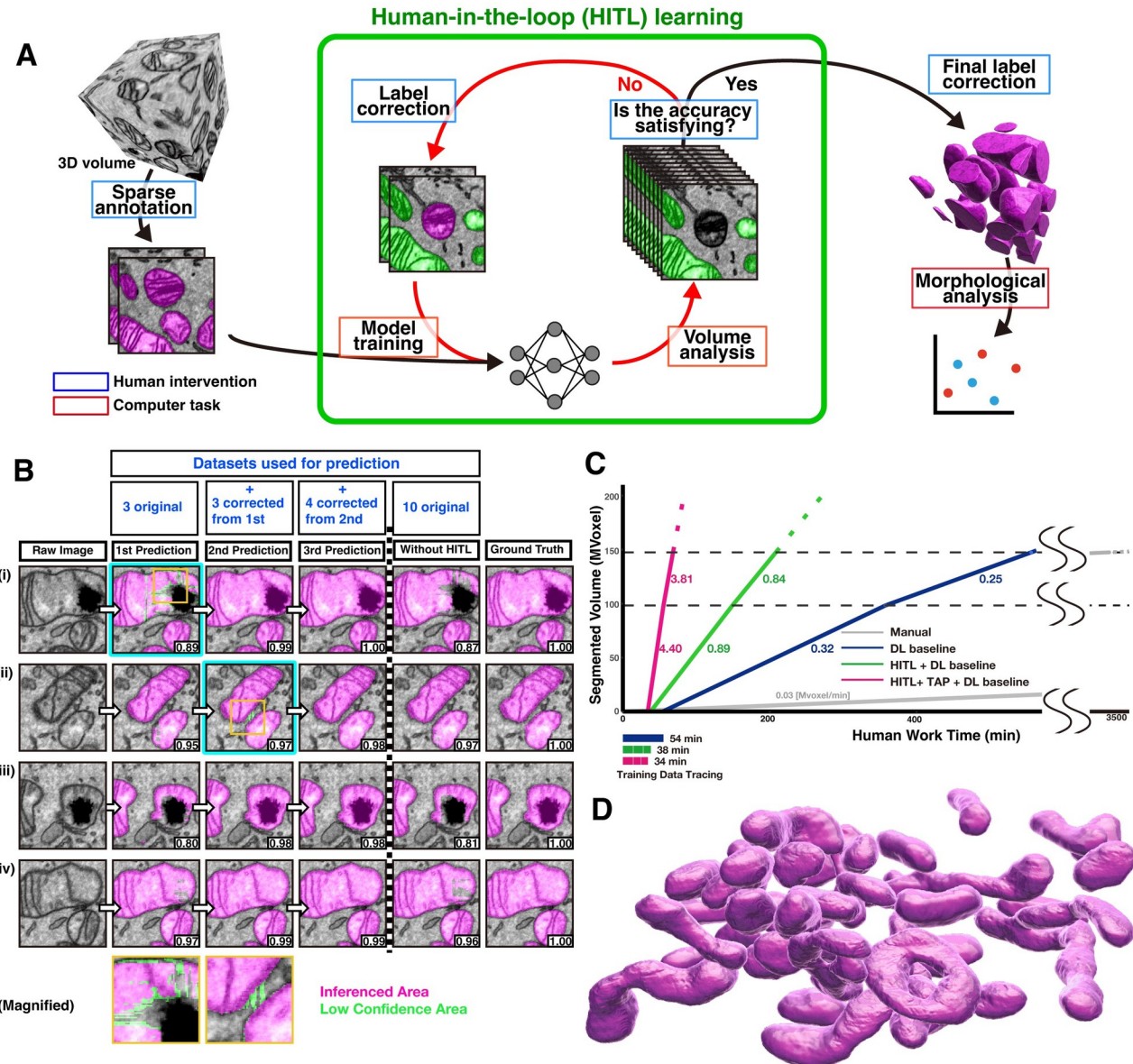

**Fig 3. The HITL-TAP method on PHILOW improved the segmentation efficiency.** (A) A diagram showing the HITL iterative workflow. The green rectangle indicates processes performed iteratively with human intervention. (B) Comparison of F1 scores between the HITL-mediated iterative learning and conventional DL. Crops of raw image only (left) or overlaid with predictions for mitochondria (magenta) are shown. The F1 score of each crop is shown at the right bottom. In the HITL-mediated iterative learning, annotations of 3 randomly picked areas were used as an initial training dataset. After the first prediction, the annotations on 3 image crops, including image (i), were corrected and combined with the initial training dataset. F1 scores of second prediction with this new training dataset were improved not only in the image (i) but also in the image (ii)–(iv). Four image crops including image (ii) were corrected after the second predictions and combined with the training dataset for the second prediction. After these cycles, the F1 scores were above 0.98 in (i)–(iv). In contrast, without HITL, even with the same number of training datasets, the F1 scores reached only 0.81–0.97. Magnifications of the areas marked with orange rectangles are shown in the bottom. Low confidence areas were highlighted in green to draw attention of the annotators for the manual correction. (C) Times required for correcting the mitochondrial prediction results obtained either inside (0–100 Mvoxel) and outside (100–150 Mvoxel) of the volume used for generating the training data by indicated methods. Magenta: HITL + TAP + DL baseline learning. Green: HITL + DL baseline learning. Dark blue: DL baseline (2D UNet++) only. Gray: without DL (Manual). The speed was calculated from the actual time required for correcting 150 Mvoxel (HITL + TAP + DL baseline, HITL + DL baseline, and DL baseline) or the time estimated from 0.3 Mvoxel of manual correction (Manual). Bars below the graph show the time required for making the training datasets. Numbers below each line indicate voxels visually inspected and corrected in 1 min (Mvoxel/minute). Source data can be found in **S3 Data**. (D) Representative 3D mitochondrial structures reconstructed from segmentations generated using HITL-TAP method on PHILOW. Scale bar, 500 nm. The raw EM data are deposited in the EMPIAR (EMPIAR-11449). EMPIAR, Electron Microscopy Public Image Archive; EM, electron microscopy; HITL, human-in-the-loop; TAP, three-axes prediction; 3D, three-dimensional; DL, deep learning; 2D, two-dimensional; PHILOW, Python-based human-in-the-loop workflow.

100 Mvoxel of 10 nm isotropic sSEM images (**Fig 3C** and **S3 Data**). Based on the time required for manual tracing of 0.9 Mvoxels, an estimated 3,546 min of human work would be required for a 100 Mvoxel manual reconstruction (**Fig 3C** and **S3 Data**). Application of a conventional DL algorithm, a 2D Unet++ model trained with randomly selected area (DL baseline), reduced the work time to 358 min (54 min for tracing of training data and 304 min of final correction). Strikingly, 3 iterative cycles of HITL (HITL + DL baseline) reduced the work time to 149 min (38 min for tracing of training data and 111 min of final correction). When the analysis was extended to neighboring image blocks, the efficiency was still better using HITL (0.84 Mvoxel/ minutes, compared to 0.25 Mvoxel/minutes in DL baseline only). A combination of TAP and HITL on DL baseline (HITL-TAP) further reduced the work time (34 min for tracing of training data and 56 min of final correction) (**Fig 3C** and **S3 Data**). The same reduction of time required for segmentation with the HITL and TAP was observed for another human annotator (**S3 Fig** and **S14 Data**). Therefore, as the database becomes larger, the total time required is 9 to 10 times shorter. These results demonstrate that PHILOW-mediated seamless 3D prediction and correction cycles significantly increased the efficiency of 3D reconstructions of mitochondria structures from sSEM images by reducing time spent on the training data generation and proofreading (**Fig 3D**).

## The HITL-TAP method enables precise segmentation of lamellar and tubular cristae

Although crista structures have been proposed to dictate functions of the mitochondria, quantitative investigation of crista structure has been mostly limited to 2D or thin sections for ET. Several previous studies manually reconstructed three-dimensional (3D) crista structures, but mostly limited to the lamellar structures or simple cristae of algae and trypanosomes from small numbers of mitochondria since manual reconstruction is highly laborious, especially for tortuous and thin tubular structure [13,14,19–21]. Therefore, equipped with the HITL-TAP on PHILOW, we tested if it can efficiently segment crista structure including tubular structures from serial FIB-SEM images. We segmented the inner structures of 24 mitochondria of a total of 1,500 μm$^3$ in 10 nm isotropic sSEM images by a single iterative HITL-TAP cycle that took a total 81 min of human work time (**Fig 4A** and **S1 Movie**). After removing objects smaller than 20 voxels as misannotations, HITL-TAP provided us with highly accurate segmentations of cristae. To examine if further proofreading by a human annotator was required, similarity between the predicted data and manual segmentations by 2 independent experts was examined by calculating F1 scores (**Fig 4B**). HITL iterations were performed on a 93 × 109 × 115 voxels volume, and a total of 7 slices out of 93 slices were used to train the model to predict all 93 slices. For lamellar cristae, F1 scores between the prediction and the manual segmentations were comparable to that between 2 manual segmentations (**Fig 4B and 4C**). Further, the precision score was even higher than those of manual segmentations (**Fig 4C**). These data suggest that the variances between the prediction and the manual segmentations are within the range of annotator-to-annotator variance. A post hoc visual inspection in 3D (from yz and zx-planes) found that the low precision score of manual segmentations was due to false-positive segmentations in marginal areas (**Figs 4D, 4E,** and **S4A–S4E**). Together with the smoother surface of the prediction as represented by smaller IoU variances in xy-planes (**Fig 4F** and **S4 Data**), we conclude that lamellar cristae segmentation by HITL-TAP method has superior reliability compared to human experts. Of note, the prediction by the HITL-TAP method included a significant portion of tubular cristae that both human experts missed, as represented by the low recall scores (0.399 and 0.539) of the manual segmentations against the prediction (**Fig 4C**). The post hoc 3D visual inspection confirmed that segmentation of tubular

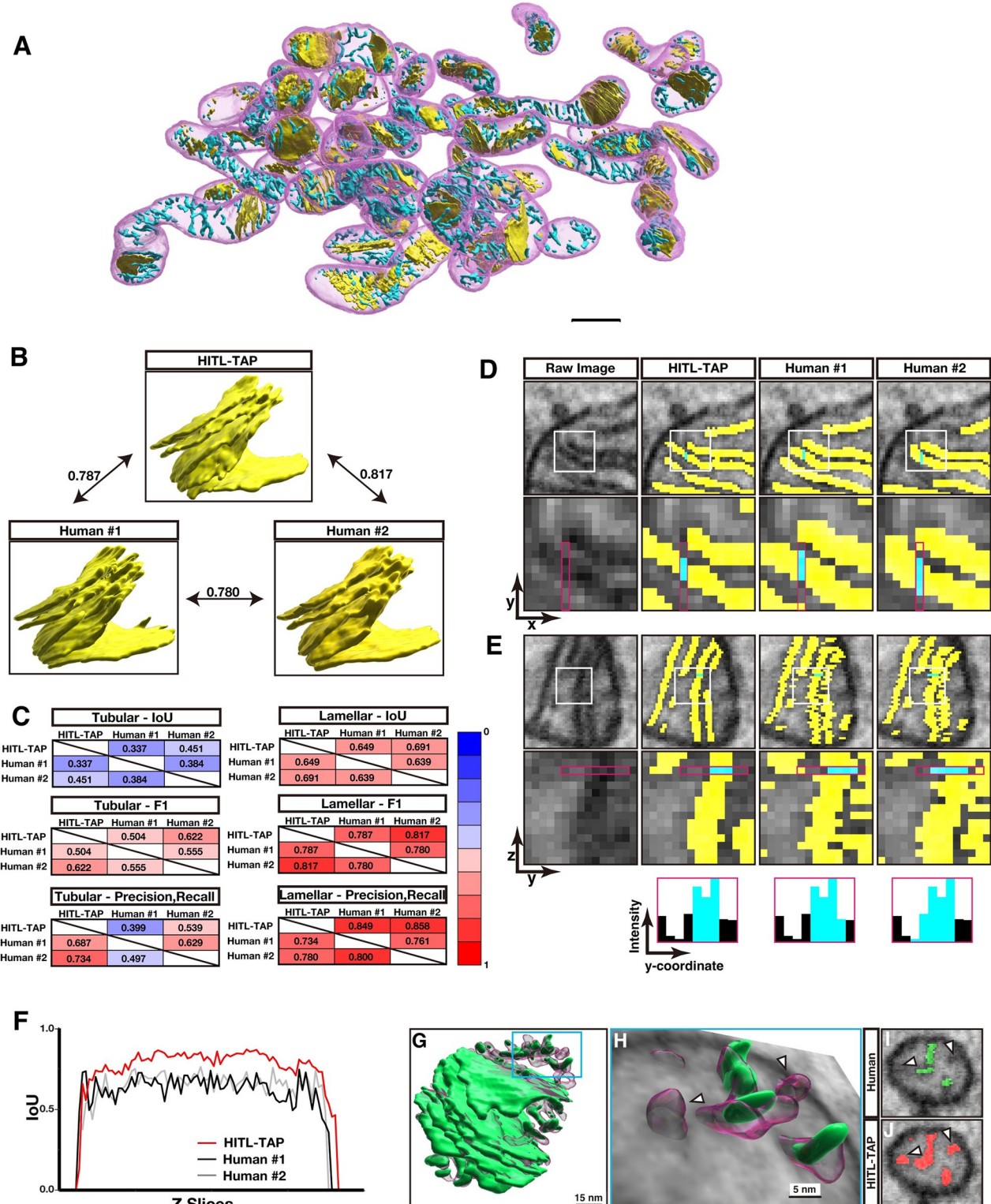

**Fig 4. Prediction of crista structures with superhuman accuracy.** (A) Representative crista structures in the mitochondria shown in **Fig 3D**.
Yellow: lamellar structure, Cyan: tubular structure. Scale bar, 500 nm. Three random images of 24 mitochondria were prepared as initial training data
for both lamellar and tubular cristae prediction. (B) Comparison of F1 scores among an HITL-TAP prediction and annotations by 2 human experts.
Corresponding 3D reconstructed lamellar images are shown. The F1 scores between the HITL-TAP prediction and the annotations by human
experts #1 and #2 are 0.787 and 0.817, respectively. This value is higher than the score between human experts (0.780). (C) IoU, F1 score, and

precision/recall on tubular structures or lamellar structures between the prediction by the HITL-TAP algorithm and one of the 2 annotations by human experts. **(D, E)** Segmentation of lamellar structures by HITL-TAP algorithm and human annotators shown in xy-planes (D) and yz-planes (E). The lower panels show areas indicated by rectangles in the upper panels. Yellow: segmentations of lamellar structures. Note that segmentations by human annotators are broader. The narrower HITL-TAP segmentation is more accurate at the voxels highlighted with cyan judging from the continuity in the z-axis (E). **(F)** Z-axis continuities of lamellar structures segmented by HITL-TAP algorithm (red), human annotator #1 (black) and human annotator #2 (gray) were indicated by IoU between neighboring xy-planes. Source data can be found in **S4 Data**. **(G)** Segmentation of crista structures by HITL-TAP algorithm (transparent magenta) and a human annotator (green). The rectangle shows the area shown in (H). **(H)** Reconstruction of tubular cristae is shown with a slice of serial EM images. Note that the human annotation missed the structures HITL-TAP algorithm segmented (arrowheads). **(I, J)** yz-view EM images corresponding to the slice in (H) annotated by a human annotator (I) and HITL-TAP algorithm (J). Arrowheads are corresponding to those pointing the tubular structures in (H). The raw EM data are deposited in the EMPIAR (EMPIAR-11449). EMPIAR, Electron Microscopy Public Image Archive; HITL, human-in-the-loop; TAP, three-axes prediction; 3D, three-dimensional; IoU, intersection over union; EM, electron microscopy.

cristae was indeed more efficient than manual segmentation (**Figs 4G–4J and S4B and S4C**, arrow heads **and S2 Movie**). These results show that our HITL-TAP method achieved super-human accuracy in segmenting both lamellar and tubular crista structures and did not require further human proofreading.

## 3D reconstruction of crista junctions from FIB-SEM images

The regulation of the crista junction (CJ), a narrow attachment point of a crista to the inner boundary membrane (IBM), is critical for crista morphogenesis (**Fig 5A**). To investigate the structure of the CJ, methods for visualizing and quantifying it in 3D are required. First, we examined if the terminus of cristae contiguous with the mitochondrial surface in the FIB-SEM images could be defined as a CJ. Since it was reported that the distance between the outer mitochondrial membranes (OMM) and IBM becomes closer in the area juxtaposed with CJ, we tested if this is the case for the putative CJ defined as above. Although the resolution of FIB-SEM in this study was not sufficient for differentiating OMM and IBM, we defined the distance between them as the thickness of the membrane structure surrounding the mitochondria (**S5A Fig and S15 Data**). Strikingly, among randomly selected 100 putative CJs, 86% displayed a narrowing in the distance between the OMM and IBM on at least 1 side (**S5B Fig**). This suggests that the contiguity to the surface of the mitochondria is a reliable indicator for defining the CJ. Therefore, we next tested if CJ can be defined based on the HITL-TAP-based cristae segmentation. First, the terminus of cristae within 30 nm of the mitochondrial surface was detected as candidates of CJ. Then, CJs were selected from the candidates by examining their contiguity with corresponding mitochondrial surfaces (see Methods). By repeating this for CJ candidates within 40 and 50 nm of the mitochondrial surface, we determined CJs (**Fig 5B and 5C and S3 Movie**). Visual inspection of those CJs mapped on the reconstructed cristae confirmed that this method precisely and comprehensively detects CJs human annotator assigned as CJs. The magnified reconstructions reveal that CJs of lamellar cristae are elongated and linked, encompassing the considerable portion of the lamellar cristae boundary (**Fig 5C**). In contrast, the CJs of tubular cristae are situated in discrete clusters. These results indicate that the 3D reconstruction of FIB-SEM images by the HITL-TAP method is applicable for efficient detection of CJs.

## OPA1 maintains a high ratio of tubular cristae

The validation of our methods indicated that previous manual segmentation of EM images missed a significant portion of tubular cristae. This might also have affected studies for revealing functions of proteins localizing at IMM, such as the dynamin-related GTPase Optic Atrophy 1 (OPA1), a gene responsible for optic atrophy [22,23]. Therefore, to determine the exact amount and structure of cristae, we extended the analysis to mitochondria expressing either

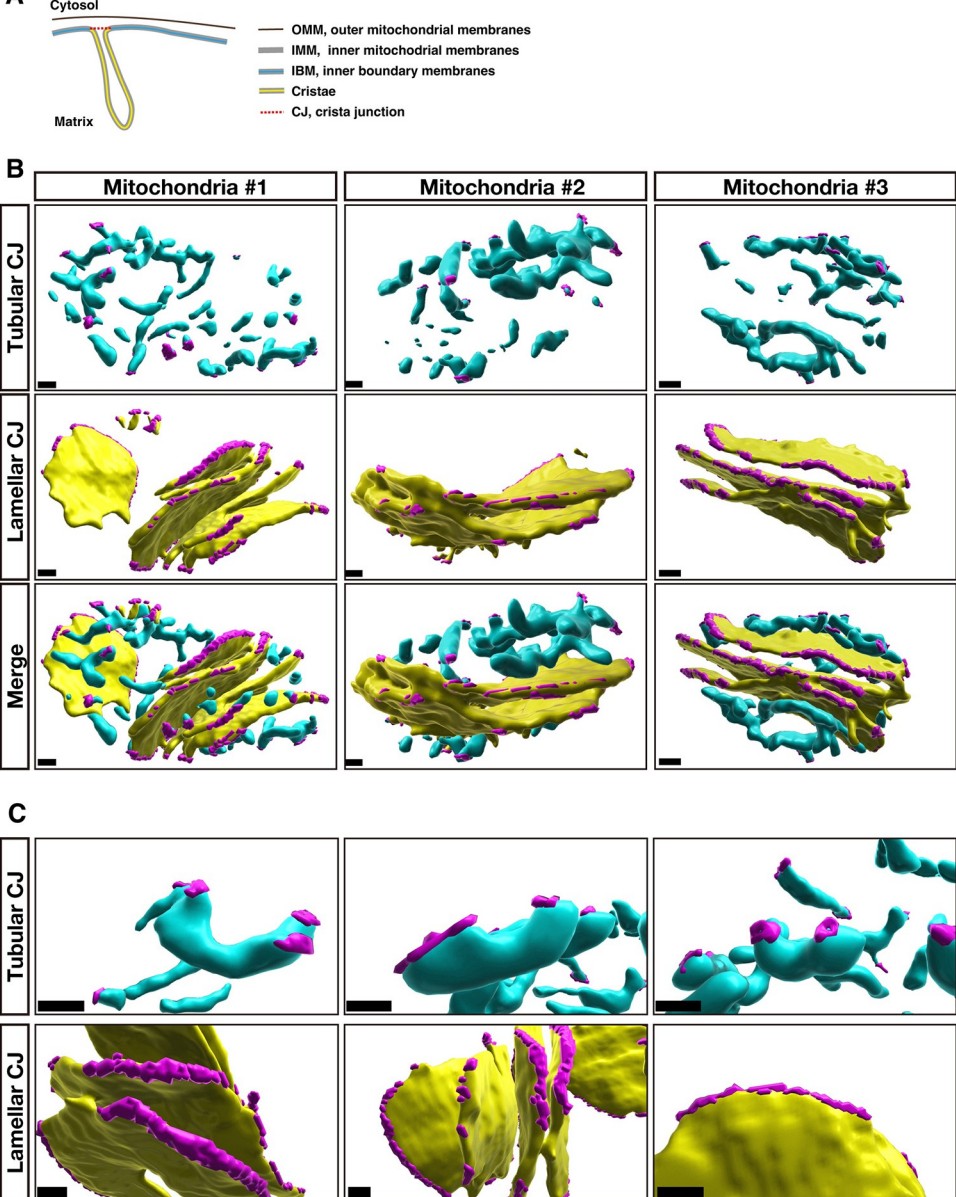

**Fig 5. 3D structure of crista junctions reconstructed from FIB-SEM images. (A)** Diagram of mitochondrial subdomains. **(B)** 3D reconstructions of CJs (magenta) overlaid with tubular (cyan) or lamellar (yellow) cristae. Scale bars, 100 nm. **(C)** Magnified images of 3D reconstructed CJs and cristae. Scale bars, 100 nm. The raw EM data are deposited in the EMPIAR (EMPIAR-11449). CJ, crista junction; EMPIAR, Electron Microscopy Public Image Archive; EM, electron microscopy; FIB-SEM, focused ion beam-scanning electron microscopy.

control shRNA (Control) or shRNA against OPA1 (OPA1 KD) from 5 cells each (**Fig 6A and 6B**; $n$ = 135 for the control; 324 for OPA1 KD, **S1, S4 and S5 Movies and S6 Fig and S16 Data**).

Three-dimensional observation of more than 100 mitochondria in control cells provided us with previously unattainable quantitative information about mitochondrial nanostructures (**S7A–S7F Fig and S17–S22 Data**, for basic properties, see **Table 1**). First, we analyzed the complexity of mitochondria by measuring mean mitochondrial surface area per volume. The

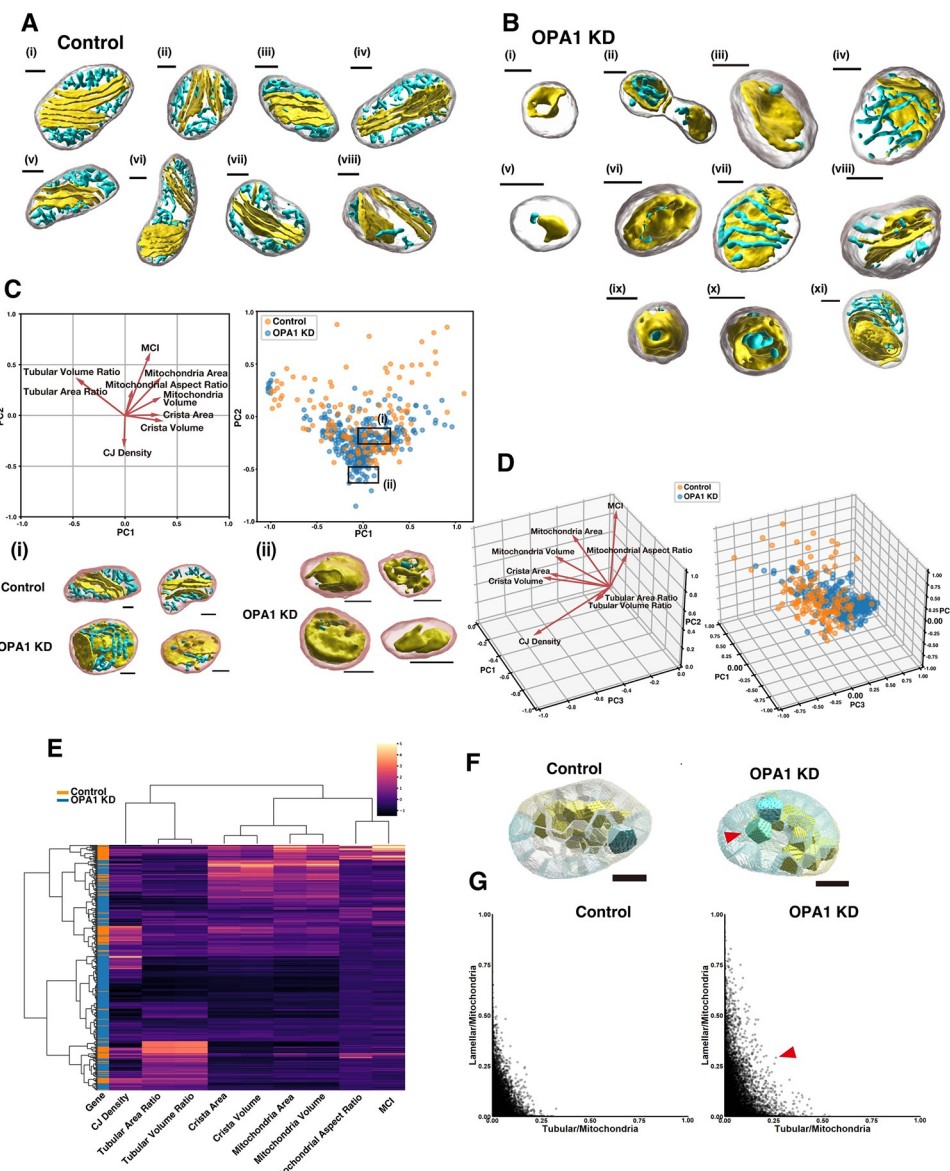

**Fig 6. Unsupervised analyses of mitochondria and crista structure in the control and OPA1 KD cells. (A, B)**
Representative crista structures of the control (**A**) or OPA1 KD (**B**) mitochondria. Scale bars, 300 nm. (**C**) PCA of
individual mitochondria. Parameters of the control mitochondria were used to fit the model. The mitochondria in the
control cells are shown as orange dots and mitochondria in OPA1 KD cells are shown as blue dots (Control *n* = 134;
OPA1 KD *n* = 324). Representative reconstructed 3D mitochondrial and crista structures in the area (i) and (ii) are
shown. Scale bars, 300 nm. Source data can be found in **S5 Data and S6 Data**. (**D**) 3D view of the PCA result shown in
(C). (**E**) Hierarchical clustering using Ward's method. Clustering revealed that there are a cluster with a high tubular
cristae ratio and enriched with control-derived mitochondria and a cluster with a low tubular cristae ratio
characteristic of OPA1 KD. Source data can be found in **S7 Data**. (**F, G**) Spatial distribution analysis of the tubular and
lamellar cristae. Transparent polygons are submitochondrial volumes divided by the K-means clustering. The ratio of
lamellar or tubular cristae in each volume is indicated by the intensity of the blue or yellow colors, respectively (F).
Note that there is a green compartment in OPA1 KD cells (arrowhead), while compartments in the control were either
yellow or blue. The horizontal axis shows the percentage of tubular cristae in mitochondrial subvolume and the vertical
axis shows the percentage of lamellar cristae (G). Each dot represents a mitochondrial subvolume. The arrowhead
indicates a dot corresponding to the green compartment in (F). Source data can be found in **S8 Data**. The raw EM data
are deposited in the EMPIAR (EMPIAR-11449). EMPIAR, Electron Microscopy Public Image Archive; EM, electron
microscopy; OPA1, optic atrophy 1; 3D, three-dimensional; PCA, principal components analysis, CJ, crista junction;
MCI, mitochondrial complexity index.

**Table 1. Mitochondrial properties in control.**

| | Volume (µm³) | Surface (µm²) | Surface/volume (µm⁻¹) | Max length (µm) | Mid length (µm) | Min length (µm) | Max length/Min length | Mid length/Min length | MCI |
|---|---|---|---|---|---|---|---|---|---|
| **Min** | 0.056 | 0.49 | 3.3 | 0.56 | 0.26 | 0.17 | 1.4 | 1.0 | 0.16 |
| **Max** | 2.9 | 19 | 9.4 | 7.6 | 2.3 | 1.0 | 30 | 6.7 | 5.0 |
| **Mean** | 0.45 | 2.4 | 5.6 | 1.7 | 0.80 | 0.51 | 4.1 | 1.6 | 0.48 |
| **Median** | 0.40 | 2.0 | 5.5 | 1.4 | 0.76 | 0.51 | 2.5 | 1.5 | 0.30 |
| **SD** | 0.32 | 1.8 | 1.2 | 0.99 | 0.30 | 0.16 | 4.1 | 0.71 | 0.51 |

MCI, mitochondrial complexity index.

obtained value was about 5.6/µm, which is similar to the value a recent study using FIB-SEM reported (6.8/µm in the liver [24]). Given that these values are smaller than the values in other cell types (8–25/µm, [25,26]), it is suggested that the evaluation of morphology highly depends on imaging techniques and sample treatments. Second, our data indicate that the surface area of cristae per mitochondria volume is 6.6/µm on average (**Table 2**). Additionally, we found that the surface/volume is similar between tubular and lamellar cristae (59/µm and 52/µm, respectively, **Table 2**). This suggests that the morphological variety of cristae is more likely to be important for functions such as controlling diffusion of ion species rather than increasing the surface area [5]. Importantly, although previous observation of mitochondrial cristae in fibroblasts showed that the lamellar structure is highly dominant [27,28], our quantitative analysis showed that the surface area of the tubular structure as a percentage of the total surface area in each mitochondrion varied widely, spanning 2% to 100% with a mean of 42% (**Table 2**). This suggests that the amount of tubular structure was vastly underestimated in previous studies.

OPA1 is required for proper cellular respiration and apoptosis [29,30]. Although studies using either 2D EM images or ET revealed that OPA1 mediates IMM fusion and the size of CJs, how those functions of OPA1 affect the arrangement of cristae is unclear [22,23,31–36]. Consistent with previous observations, our 3D analysis statistically showed that the maximum over minimum length ratio of mitochondria in OPA1 KD cells were smaller compared to the control, indicating that OPA1-deficient mitochondria were more spherical (**Table 3, S8A Fig and S23 Data**). Further, we observed that lamellar cristae were thicker, and the diameter of tubular cristae was larger in OPA1 KD compared to the control (**Table 4, S9A–S9C Fig, and S26 and S27 Data**). This result was further supported by measuring the width of randomly selected 100 CJs on the xy-plane with the original resolution of 5 × 5 nm/px (**S9D and S9E Fig and S28 Data**). These phenotypes confirm that OPA1 functions in narrowing the CJ [31,37]. To analyze mitochondrial ultrastructure systematically, given the large number of

**Table 2. Crista properties in control.**

| | Crista volume (µm³) | Crista surface (µm²) | Crista surface/ mitochondrial volume (µm⁻¹) | Tubular volume (µm³) | Tubular surface (µm²) | Tubular surface/ volume (µm⁻¹) | Lamellar volume (µm³) | Lamellar surface (µm²) | Lamellar surface/volume (µm⁻¹) | Tubular ratio (surface) |
|---|---|---|---|---|---|---|---|---|---|---|
| **Min** | 0.0012 | 0.083 | 0.32 | 0.00010 | 0.0084 | 39 | 0.00023 | 0.015 | 23 | 0.021 |
| **Max** | 0.34 | 19 | 12 | 0.095 | 6.1 | 94 | 0.24 | 13 | 82 | 1.0 |
| **Mean** | 0.062 | 3.1 | 6.6 | 0.018 | 1.0 | 59 | 0.048 | 2.3 | 52 | 0.42 |
| **Median** | 0.058 | 2.9 | 6.7 | 0.015 | 0.82 | 59 | 0.044 | 2.1 | 53 | 0.34 |
| **SD** | 0.049 | 2.5 | 2.5 | 0.014 | 0.78 | 8.7 | 0.039 | 1.9 | 11 | 0.25 |

**Table 3. Mitochondrial properties in OPA1 KD.**

| | Volume (μm³) | Surface (μm²) | Surface/volume (μm⁻¹) | Max length (μm) | Mid length (μm) | Min length (μm) | Max length/min length | Mid length/min length | MCI |
|---|---|---|---|---|---|---|---|---|---|
| **Min** | 0.0065 | 0.13 | 3.0 | 0.27 | 0.24 | 0.17 | 1.2 | 1.0 | 0.14 |
| **Max** | 1.4 | 4.7 | 20 | 2.3 | 1.6 | 1.1 | 6.7 | 5.4 | 1.0 |
| **Mean** | 0.26 | 1.4 | 6.6 | 1.0 | 0.76 | 0.50 | 2.1 | 1.5 | 0.30 |
| **Median** | 0.19 | 1.2 | 6.2 | 0.96 | 0.71 | 0.48 | 1.8 | 1.4 | 0.28 |
| **SD** | 0.21 | 0.82 | 2.2 | 0.36 | 0.28 | 0.15 | 0.83 | 0.54 | 0.11 |

MCI, mitochondrial complexity index; OPA1, optic atrophy 1.

ultrastructural segmentation of mitochondria, we performed unsupervised learning of 3D morphological features of mitochondria and cristae. We used a nine-parametric (mitochondrial volume, mitochondrial area, mitochondrial aspect ratio, mitochondrial complexity index (MCI) [38], crista volume, crista area, tubular volume ratio, tubular area ratio, and CJ density) principal component analysis (PCA) to categorize mitochondrial ultrastructural features in an unbiased way (**Fig 6C and 6D**, **Tables 1–4, and S5 and S6 Data**). The parameters of mitochondria in control cells were used to fit the model, and the resulting model was subjected to dimensionality reduction algorithms for all mitochondria, including those in OPA1 KD cells. The OPA1 KD mitochondria form a distinct cluster with a PC2 score lower than −0.5, which enables an unbiased identification of OPA1 KD mitochondria only from structural information (**Fig 6C**). We then performed hierarchical clustering of mitochondria using the Ward's method (**Fig 6E and S7 Data**). Mitochondria were largely divided by the amount of volume and surface area. Interestingly, within the smaller mitochondria cluster, there was a clearly delineated cluster with high tubular ratio, where the control mitochondria were enriched. In addition, there was a cluster with a very small tubular ratio, most of which were OPA1 KD mitochondria. These results suggest that differences in the ratio of lamellar-tubular cristae best represent the phenotype of OPA1 KD, rather than mitochondrial size or density of cristae (**S8A–S8I Fig**). Indeed, examination of crista structure revealed that the ratio of tubular structures was significantly lower in OPA1 KD mitochondria (mean 25%, $p < 0.0001$, **S8I Fig**). Importantly, consistent with the fact that the surface/volume ratio is comparable between tubular and lamellar cristae, changes in the ratio between tubular and lamellar structure in OPA1 KD cells did not affect the total surface area of cristae (**S8G Fig**). This indicates that OPA1 determines the ratio of tubular versus lamellar structures while leaving the crista area constant.

**Table 4. Crista properties in OPA1 KD.**

| | Crista volume (μm³) | Crista surface (μm²) | Crista surface/ mitochondrial volume (μm⁻¹) | Tubular volume (μm³) | Tubular surface (μm²) | Tubular surface/ volume (μm⁻¹) | Lamellar volume (μm³) | Lamellar surface (μm²) | Lamellar surface/volume (μm⁻¹) | Tubular ratio (surface) |
|---|---|---|---|---|---|---|---|---|---|---|
| **Min** | 0.0 | 0.0 | 0.0 | 0.000023 | 0.0020 | 34 | 0.00022 | 0.017 | 19 | 0.0 |
| **Max** | 0.23 | 11 | 13 | 0.056 | 2.6 | 100 | 0.21 | 9.1 | 87 | 1.0 |
| **Mean** | 0.044 | 1.9 | 6.3 | 0.0094 | 0.47 | 55 | 0.038 | 1.5 | 45 | 0.25 |
| **Median** | 0.029 | 1.3 | 6.6 | 0.0064 | 0.32 | 54 | 0.026 | 1.0 | 45 | 0.21 |
| **SD** | 0.044 | 1.8 | 2.7 | 0.0099 | 0.48 | 11 | 0.037 | 1.4 | 11 | 0.20 |

OPA1, optic atrophy 1.

We also investigated spatial arrangement of the lamellar and tubular cristae. Each mitochondrion was divided into around $5 \times 10^{-3}$ μm$^3$ subvolumes by a K-means non-hierarchical clustering method (**Fig 6F**). Calculation of the tubular or lamellar cristae ratio in each subvolume revealed that either tubular or lamellar structure represents more than 80% of cristae in all the cristae containing subvolumes (**Fig 6G** and **S8 Data**). This suggests that lamellar and tubular cristae are well segregated in the intramitochondrial spaces of control cells. Interestingly, this segregation was compromised in OPA1 KD mitochondria. This suggests that OPA1 is required for the spatial allocation of tubular and lamellar cristae.

## OPA1 controls the orientation of lamellar cristae

We next examined the orientation of lamellar cristae relative to mitochondrial shape. The angle between the longitudinal direction of the mitochondria (mitochondrial orientation) and the direction perpendicular to the plane of lamellar cristae in 3D was measured. Therefore, if the angle is zero, the lamellar cristae are perpendicular to the mitochondrial orientation. Although we observed lamellar cristae that clearly aligned perpendicular to the mitochondrial orientation, as previously observed with super resolution microscopy ([39–42], **Fig 6A(i), (iii), and (vi)**), a significant number of cristae were aligned parallel to mitochondria orientation (**Fig 6A(iv) and (vii)**). Indeed, the average angle was around 60˚ in the control (**Fig 7A and 7B**, **S9** and **S10 Data**). This suggests that crista orientation is relatively parallel to the mitochondrial orientation in control mitochondria. Furthermore, compared to the control, the alignment of OPA1 KD cristae was significantly more parallel to the mitochondrial orientation (**Figs 6B(vi) and (viii)** and **7B,** average 74˚). This finding statistically and quantitatively supports a previous 2D qualitative observation of the crista angle in the OPA1 KD mitochondria [43].

## Validation of previously proposed OPA1 deficient crista structures

The yeast *Saccharomyces cerevisiae* containing a temperature-sensitive mutation of Mgm1p, a homolog of OPA1, forms septum structures that divide mitochondria into 2 compartments at non-permissive temperatures [35]. A recent study also suggests that the number of septum-like cut-through crista is increased in OPA1 KO MEF [44]. However, to our knowledge, a complete septum structure has never been shown. Creating a septa-detecting algorithm (see Methods section) and following visual inspection, we found 4 complete septa among 324 OPA1 KD mitochondria (**Fig 7C**). Since none of the 135 control mitochondria possessed a septum, our data show that OPA1 KD may induce septum formation as previously predicted.

A stereological analysis of a curved septum suggested that OPA1 KD also produces mitochondria with an onion-like structure [35,44]. Indeed, we observed onion-like structures in 2D sections of OPA1 KD mitochondria (**Fig 7D**). Although FIB-SEM analysis of OPA1 KO MEF and tomographic reconstruction of *C. elegans* harboring a mutation in the GTPase domain of the OPA1 homolog eat-3 suggested that the onion-like cristae membranes are detached from IBM, a whole mitochondrial structure of such cristae was not observed [44,45]. Our whole-mitochondrial reconstruction of those structures revealed that they are part of cristae but mostly disconnected from the IBM and folding inside of the mitochondria (**Fig 7E** and **S6 Movie**).

Finally, we examined the role of OPA1 in regulating CJ density. Counting the number of CJs per cristae surface area showed that OPA1 KD had no significant effect on the density of CJs associated with lamellar cristae, but significantly decreased it for the tubular cristae compared to the control (**Fig 7F–7H** and **S11–S13 Data**). This finding emphasizes the advantage of our comprehensive 3D analysis of cellular ultrastructure to identify new crista structure

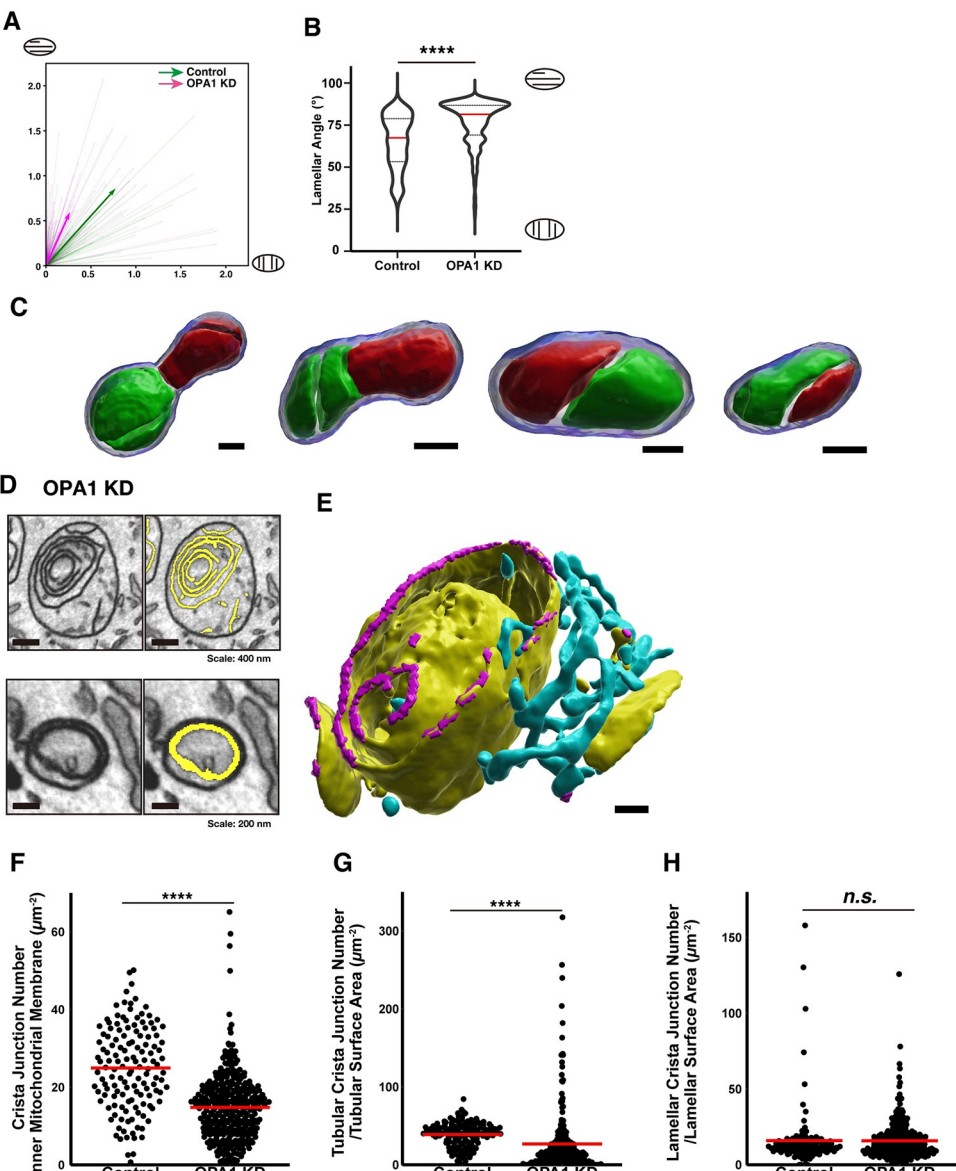

**Fig 7. Phenotypes for OPA1-deficient crista structures.** (**A**) The light colored vectors indicate the angular direction between the vector perpendicular to each lamellar cristae and the mitochondrial long-axis vector. The magnitude was normalized by the inverse of the eigenvalue. The dark colored vectors are the average of vectors from all mitochondria scaled by a factor of 4. Source data can be found in **S9 Data**. Green: Control; Magenta: OPA1 KD. (**B**) The average of the angles was calculated for each mitochondrion. Note that the angles are diverse in the control, while most of the angles are concentrated close to 90° in OPA1 KD. Source data can be found in **S10 Data**. ****$p < 0.0001$, Mann–Whitney test. (**C**) Four OPA1 KD mitochondria with complete septa. Each mitochondrion was separated into the green and red compartments by inner membranes. These compartmentalized mitochondria were defined by a septa-detecting algorithm. Scale bars, 20 nm. (**D**) Representative EM images of onion-like structures in the OPA1 KD cells. Yellow: segmentations of lamellar structures. (**E**) 3D reconstruction of the onion-like crista structure shown in (D). Yellow: lamellar cristae; Cyan: tubular cristae; Magenta: CJ. Scale bar, 150 nm. (**F–H**) Statistical analysis of the CJ density in the control and OPA1 KD. CJ density is calculated by dividing the number of CJ by the crista surface area (F). Tubular CJ density is calculated by dividing the number of tubular crista by the tubular surface area, and lamellar CJ density is calculated in the same way (G, H). Source data can be found in **S11–S13 Data**. ****$p < 0.0001$, Mann–Whitney test. The raw EM data are deposited in the EMPIAR (EMPIAR-11449). EMPIAR, Electron Microscopy Public Image Archive; OPA1, optic atrophy 1; EM, electron microscopy; CJ, crista junction.

phenotypes and more generally improve our understanding of the molecular mechanisms regulating the structure/function relationship of mitochondrial physiology.

## Discussion

Since the first description of mitochondria ultrastructural features, cristae have been intensely studied as a prominent structural feature of mitochondria. However, how this uniquely convoluted ultrastructure contributes to biochemical reactions occurring at mitochondria still remains a subject of intense debate. One of the main roadblocks for analyzing the relationship between ultrastructure and functional properties of cristae is the lack of technology to quantify these nanostructures in a micrometer-scale context. In this study, we quantitatively analyzed crista structures in hundreds of complete mitochondria by implementing an HITL-TAP method on analyzing sSEM images. This was enabled by engineering a DL image analysis platform PHILOW. Although DL algorithms, isotropic FIB-SEM imaging, and high-performance computers have been developed individually, it is only the efficient combination of these in our human intervention interface by PHILOW that allows us to observe ultrastructures that humans cannot readily recognize. The analyses provided us with precise basic properties of crista structure and revealed that tubular cristae are more abundant than previously reported. Application of the HITL-TAP analysis and subsequent unbiased PCA revealed that an optic atrophy-related gene OPA1 regulates the balance between tubular and lamellar cristae of mitochondria. As such, our method provides ultrastructural information and insights into the regulatory machinery of cristae to reveal the structure–function relationship of the inner workings of mitochondria.

### Implementation of highly precise DL-based segmentation using PHILOW

We built PHILOW on top of napari, a multidimensional image viewer for python, equipped with a GUI and workflows suitable for handling 3D data. Therefore, PHILOW enabled us to perform the HITL-TAP method on FIB-SEM images, which accelerated the segmentation speed by approximately 15 times and reached superhuman accuracy in segmenting crista structures. Although the HITL algorithm has been proposed as an efficient strategy for integrating human knowledge and DL, its application has been limited. The existing tools for iterative segmentations are either based on classical machine learning techniques such as random forest [46] or designed for segmenting 2D images [47]. Our study provides a proper platform to utilize HITL for improved efficiency and accuracy of DL. Our detailed evaluation of DL-based prediction by comparison with manual segmentation revealed that the visual recognition of tubular cristae from a single plane is a particularly error-prone process. In addition, the inconsistency in determining boundaries of tortuous objects by human annotators resulted in coarse surfaces of reconstructed cristae. Thus, our 3D analysis using HITL-TAP reaches superhuman accuracy.

To perform segmentation on 3D volume datasets of EM and fluorescence microscope images, DL algorithms using information from multiple slices, such as 2.5D U-Net [48] and 3D U-Net [49–52], have been developed and various types of semantic segmentation such as organelles have been investigated. However, creating 3D annotated training data is a very labor-intensive work, as it involves repeatedly annotating tens to hundreds of consecutive slices. While the HITL-TAP on PHILOW and 2D Unet++ has the advantage of quickly creating a high-performing model with fewer slices (10 or less slices in most cases), PHILOW can in principle be combined with any other DL algorithm. Therefore, once a certain volume of proofreading is completed, it can be combined with 2.5D and 3D U-Net to achieve even higher performance and contribute to the reconstruction of complex 3D structures like the ER or

Golgi apparatus. The method developed in this study can be applied not only to any EM image analysis, but also to other types of image segmentation and detection.

## Basic properties of crista structure revealed by 3D analysis

With the highly accurate and high-throughput 3D reconstructions, we determined the basic numbers of cristae properties, such as entire area or number of CJs per mitochondria. Quantitative analysis of 135 control mitochondria from 5 cells provided us with reliable properties of mitochondrial structure with robust statistical sampling. The cristae areas per mitochondria volume obtained in this study were lower than the values previously proposed by estimation from analyses of partial slices of mitochondria [53]. This is attributable to the difference in the sample of origin (cell lines or tissue) and estimations without accounting for cristae-free spaces. In addition, to our knowledge, this is the first report of CJ numbers and frequency. Despite previous observations using ET indicating that mammalian CJs are predominantly tubular, our reconstruction of CJs for lamellar cristae revealed significant portions of elongated CJs in mouse fibroblasts ([9], **S10 Fig** and **S7 Movie**). Consistent with this, human fibroblasts are a rare cell type that exhibits both tubular and slot-like CJs, even under investigation by ET [54]. However, it is important to note that the resolution of FIB-SEM still might not be sufficient to distinguish 2 axially juxtaposed CJs. Therefore, further studies are necessary to examine the actual CJ structures in various cell type, particularly utilizing techniques with higher resolution, such as cryo tomography. Our data also addresses the orientation of aligned cristae against mitochondrial orientation. Isolation of lamellar cristae by reconstructing complete cristae provides a way to assess the alignment of cristae even in mitochondria with a large portion of tubular cristae. In contrast to cristae orientations in HeLa cells, in the fibroblast cell line we investigate in this study, the orientation of lamellar cristae was parallel to that of mitochondria. Since chemical fixation adds artifacts to the cellular ultrastructure, further study using cells fixed with high-pressure freezing is required for a more precise evaluation of crista structures.

## OPA1 promotes tubular cristae at the expense of lamellar cristae

Segregating tubular and lamellar structure in the 3D cristae analysis revealed that OPA1 is required for the formation of tubular cristae without affecting total cristae surface area. The correlation analyses suggest that the decrease in tubular ratio in OPA1 KD is independent of mitochondrial size, mitochondrial aspect ratio, and MCI (see **S11 Fig** and **S29–S31 Data**). In addition, investigation of more than 300 whole OPA1 KD mitochondria structures identified rare septum structures and folding cristae disconnected from the IBM. These results indicate that a large part of OPA1 KD mitochondria show deficits in CJs and only a small part of them show IMM fusion-related phenotype. This is in agreement with the fact that an OPA1 mutant lacking GTPase activity, which is required for IMM fusion, can largely rescue malformation of cristae observed in OPA1 KO [55]. In line with this, the onion-like folding crista structure is observed in the cells deficient for Mic60 or Mic19, which are required for CJ formation [14].

A mathematical model suggested that the morphology of cristae determines the diffusion patterns of ions, metabolites, and proteins [5,6]. Given the heterogeneity of membrane potential among cristae in the same mitochondrion [39], simulations using actual 3D structures obtained in this study, instead of uniform hypothetical crista structure, will contribute to revealing roles of inner mitochondrial structure in regulating physiological functions of mitochondria, such as apoptosis [30]. Considering that OPA1 compartmentalizes the membrane potential between cristae and IBM [39], incorporating the cristae reconstructions from OPA1 KD mitochondria in simulating proton diffusion will be of great interest.

Taken together, PHILOW paves the way for unbiased large-scale analyses of the cellular ultrastructure with vastly improved accuracy and efficiency compared to previous approaches. Additionally, PHILOW allows for identification of cellular nanostructures that are not yet recognizable by humans. This pioneering technology will advance not only the field of cell biology, but also allow for capturing subtle phenotypic symptoms occurring in the early stages of diseases at the ultrastructural level of vital organelles.

## Methods

### Cell culture

NIH3T3 (BRC, RCB2767) cells were maintained with Dulbecco's Modified Eagle Medium (DMEM, Sigma-Aldrich, catalog no. D6429) supplemented with 10% FBS (Biowest, catalog no. S1760-500), 1% Penicillin-Streptomycin (Gibco, catalog no. 15140–122), and 5 μg/ml Plasmocin prophylactic (Invitrogen, catalog no. ant-mpp) at 37°C under 5% $CO_2$.

### DNA plasmids

pH1-shControl-pSyn-tdtomato and pH1-shOPA1-pSyn-tdtomato were generated from pH1-shSRGAP2-pSyn-tdtomato (kind gift from Dr. Franck Polleux, generated from pSCV2 [56]) by replacing the shSRGAP2 sequence with hybridized oligomers of shControl (5′- TCG AGC CGC AGG TAT GCA CGC GTT CAA GAG ACG CGT GCA TAC CTG CGG TTT TTG T -3′) and shOPA1 (5′- CCG GCA TGG AAG AAG AAC CAT ATT TCT CGA GAA ATA TGG TTC TTC TTC CAT GTT TTT G -3′), respectively, using XhoI and XbaI sites.

### FIB-SEM

NIH3T3 cells infected with lentivirus carrying either the control or shOPA1 were fixed with 2.5% glutaraldehyde (Electron Microscopy Sciences, catalog no. 16620-P) in DMEM, for 1 h at room temperature. After washing with 0.1 M phosphate buffer (0.02 M Sodium Dihydrogenphosphate Dihydrate, 0.08 M Disodium Hydrogenphosphate), cells were scraped and collected with 0.2% BSA/0.1 M phosphate buffer followed by centrifugation at 1,450×g. The samples were post-fixed with 1% $OsO_4$ (Electron Microscopy Sciences, catalog no. 19150), 1.5% potassium ferricyanide (Fujifilm Wako Pure Chemical Corporation, catalog no. 167–03722) in a 0.05 M phosphate buffer for 2 h or 30 min. After being rinsed 3 times with $H_2O$, cells were stained with 1% thiocarbohydrazide (Sigma-Aldrich, catalog no. 223220) for 5 min. After rinsing with $H_2O$ 3 times, cells were stained with 1% $OsO_4$ in $H_2O$ for 30 min. After rinsing with $H_2O$ 2 times at room temperature and 3 times with $H_2O$ at 50°C, cells were treated with 0.635% lead nitrate (Sigma-Aldrich, catalog no. 203580), 0.4% aspartic acid (Sigma-Aldrich, catalog no. A9256), pH 5.2 at 50°C for 20 min. The final cell pellet was embedded into 2% low melting agarose (MP Biomedicals, catalog no. AGAL0050). After gel setting on ice, the embedded cell pellets were cut into small fragments (1 to 2 mm$^3$). The fragments were followed by incubations in an ascending ethanol series (10 min each in 50%, 70%, 90%, 95% ethanol/$H_2O$), 10 min in 100% ethanol 4 times. This was followed by infiltration in graded concentrations of Durcupan resin (Durcupan-ethanol for 1 day at a 1:3 dilution, 1 day at a 1:1 dilution, and 1 day at a 3:1 dilution). After incubating with 100% Durcupan resin overnight, curing of the resin was achieved at 65°C for 3 days. Durcupan resin were made by mixing 12.4 g of component A (Sigma-Aldrich, catalog no. 44611), 9.9 g of component B (Sigma-Aldrich, catalog no. 44612), 0.2 g of LUVEAK-DMP-30 (Nacalai Tesque, catalog no. 14425–62), and 0.1 ml of component D (Sigma-Aldrich, catalog no. 44614). Resin blocks were trimmed with a Trim-Tool diamond knife (Trim 45; DiATOME) in a Leica Ultramicrotome (UC7). The resin blocks

were clued with silver paint (Electron Microscopy Sciences, catalog no. 12642–14) on an SEM stub and coated with gold or platinum to guarantee electrical conductivity. The stubs were fixed on a 45˚ multi holder (Electron Microscopy Sciences) and mounted in a FIB-SEM (Helios 650, Thermo Fisher). Milling was done at 52˚, parallel to the gallium ion beam (30 kV, 770 pA), removing 10 nm material per step. Imaging was done normal to the electron beam, as described previously [57] at 1.5 keV to 2 keV, 800 pA beam current, 6,144 × 4,096 frame size, 2.7 mm to 2.8 mm working distance, 30 μm horizontal field of view, and 6 μs dwell time, using the through-the-lens detector (TLD) in backscattered electron (BSE) mode. The final voxel size was $4.88 \times 4.88 \times 10$ nm$^3$.

## Lentivirus production

Recombinant lentiviruses were produced as previously reported [58], and 293T (BRC, RCB2202) cells were co-transfected with shuttle vectors, LP1, LP2, and VSV-G using FuGENE transfection reagent (Promega, catalog no. E2311). Approximately 24 h after transfection, the media were exchanged with fresh DMEM supplemented with 10% FBS, 1% Penicillin-Streptomycin, and 5 μg/ml Plasmocin prophylactic, and 24 h later, supernatants were harvested, spun at 500×g to remove debris and filtered through a 0.45 μm filter (Sartorius). The filtered supernatant was concentrated to 100 μl using an Amicon Ultra-15 (molecular weight cut-off 100 kDa) centrifugal filter device (Merck Millipore), which was centrifuged at 4,000×g for 60 min at 4˚C. The concentrated samples (100 μl) were diluted with 150 μl of PBS and stored at −80˚C in 50 μl aliquots.

## Quantitative RT-PCR

Total RNA was isolated from NIH3T3 cells infected with lentivirus carrying either the control or shOPA1 7 days after infection, with the use of RNAiso plus (TaKaRa, catalog no. 9109), and 0.5 μg of the RNA were subjected to reverse transcription with ReverTra Ace qPCR RT Master Mix with gDNA Remover (TOYOBO, catalog no. FSQ-301). The resulting cDNA was subjected to real-time PCR with Thunderbird SYBR qPCR mix (TOYOBO, catalog no. QPS-201) in a LightCycler 96 (Roche) using the following primers [59,60]:

Actin
5′- GGCTGTATTCCCCTCCATCG -3′
5′- CCAGTTGGTAACAATGCCATGT -3′
OPA1
5′- AAGTGACAAGCATTACAGG -3′
5′- CTCCAAGATCCTCTGATACT -3′

## Image preprocessing

Cropped images were aligned using the Linear Stack Alignment with scale-invariant feature transform (SIFT) plugin, implemented in Fiji (NIH). The algorithm was run 3 times with the following parameters: (1) 1.6 pixels Gaussian blur, 3 steps per scale octave, 64 pixels minimum image size, 1,024 pixels maximum image size, 4 feature descriptor size, 8 feature descriptor orientation bins, 0.92 closest/next closet ratio, 25 pixels maximal alignment error, 0.05 inlier ratio, rigid transform and with interpolation; (2) 1.6 pixels Gaussian blur, 3 steps per scale octave, 256 pixels minimum image size, 1,024 pixels maximum image size, 4 feature descriptor size, 8 feature descriptor orientation bins, 0.92 closest/next closet ratio, 25 pixels maximal alignment error, 0.05 inlier ratio, rigid transform and with interpolation; (3) 1.6 pixels Gaussian blur, 3 steps per scale octave, 256 pixels minimum image size, 1,024 pixels maximum image size, 4 feature descriptor size, 8 feature descriptor orientation bins, 0.92 closest/next

closet ratio, 25 pixels maximal alignment error, 0.05 inlier ratio, affine transform and with interpolation.

## Machine learning

In order to segment mitochondria, we generated UNet++ models trained with the RMSprop optimizer and binary cross-entropy dice coefficient loss (BCE-dice-loss). The images for generating the training data were cropped into $512 \times 512$ pixels and subjected to the model training with 400 epochs, the batch size to 4 and the learning rate to $10^{-4}$. After the training, the model was applied to the whole image stack to predict the mitochondrial area and the mitochondrial probability of each voxel was calculated in the range of [0, 1] and lineary expanded to [0, 255]. Since predictions at the edges of images are relatively inaccurate due to the limited information from surrounding structures, the areas 10% from the top, bottom, left, and right were excluded from the evaluation of the prediction results. For the TAP method, the probabilities obtained from predictions in the xy, yz, and zx-planes were averaged for each voxel. Voxels with values exceeding 127 were then annotated as mitochondria, and voxels with [1, 126] values were indicated as low confidence. Validation was not performed during training. In measuring the F1 score at the test time, it was calculated on slices other than the slice used for the training. For segmenting lamellar and tubular cristae, UNet++ models were trained with the same hyperparameter as for the mitochondrial segmentation. To prevent misannotation at the gaps between lamellar structures, we defined "gap" as non-cristae pixels at a distance of 60 nm from the cristae and used 3 channels, lamellar cristae, tubular cristae, "gap" for the training. After non-mitochondrial areas were masked, images were applied to the models. The prediction results for tubular and lamellar cristae were processed in the same manner as for the mitochondrial prediction using the TAP method.

## Proofreading

In proofreading the mitochondria, all slices were manually checked and any errors in the predicted results were corrected to make the final results. In an experiment to examine the relationship between F1 score and proofreading time, 12 subvolumes of $100 \times 512 \times 512$ voxels were cropped from the volume for which mitochondrial reconstruction had already been completed and reanalyzed. The F1 score was calculated by comparing the predicted results of the model to the GT using the slices excluding the slices used for the training. In reconstructing crista structure, the HITL-TAP provided us with highly accurate segmentations of crista without human proofreading. Therefore, the predictions of the model developed after running several HITL iterations were not corrected by humans, but only checked by 2 experts to make sure that there were no major errors, which were used as the final result.

## Calculation and subsequent analysis of mitochondria and cristae ultrastructural features

First, we labeled mitochondria and counted labeled mitochondrial volume. Then, we extracted 1 pixel of each mitochondrial outline and counted them. Using the mitochondrial volume and surface, we calculated the MCI to assess mitochondrial morphological complexity [38] and defined the mitochondrial aspect ratio as the mitochondrial maximum length divided by the minimum length. Next, we calculated the surface area and volume of the cristae in the labeled mitochondria using the same method as for mitochondria. Finally, we calculated the surface area and volume of lamellar and tubular structures by separating whether the calculated cristae were lamellar or tubular. To calculate the angle between mitochondria and lamellar cristae, we calculated the first principal component (PC) of their voxel locations and set its axis as the

direction of the mitochondrial longitudinal axis (mitochondrial orientation). For lamellar cristae, lamellar cristae volumes are eroded at first to separate which are in contact with each other. Then, we calculated the third PC and set it as the axis perpendicular to the lamellar cristae plane. The angle between the 2 axes was then calculated. More details with the source code and documentation are available at: https://github.com/neurobiology-ut/PHILOW_Data_Manuscript.

### Variance of IoU between neighboring xy-slices

The IoU value was calculated between a given section and subsequent section. Five IoU values were selected starting from the first segmented slice, and their variance was calculated. For the manual segmentation data (Human), the IoU variance was calculated for the IoU values of pairs [2,6], and for the DL baseline and TAP + DL baseline, the IoU variances were calculated for the IoU values of pairs [3,7]. More details with the source code and documentation are available at: https://github.com/neurobiology-ut/PHILOW_Data_Manuscript.

### Detection and quantification of crista junctions

The mitochondrial segmentation was eroded by 3 px and the voxels at the surface of the eroded mitochondria overlapping with the cristae were identified as "CJ candidates." Subsequently, a direct line was established between each CJ candidate and the nearest surface of the original mitochondrial segmentation. Then, intensities along this line were quantified for each voxel. If intensities of all pixels along the line were [0, 126], that CJ candidate was defined as CJ. These procedures (CJ detection) were repeated for mitochondrial segmentations eroded by 4px and 5px, in this order. If a nearest voxel of the mitochondrial surface from a detected CJ has already been paired with another CJ in a previous detection cycle, that CJ was disregarded. Finally, among the CJs, those with smaller than 2 voxels were dismissed as erroneous extractions. The number of detected CJs was counted and normalized by the area of the IMM to calculate a CJ density. More details with the source code and documentation are available at: https://github.com/neurobiology-ut/PHILOW_Data_Manuscript.

### Unsupervised analysis of mitochondria and crista structures

For unsupervised analysis of 3D mitochondria and crista structures, mitochondria volume, mitochondria surface, mitochondrial aspect ratio, MCI, crista volume, crista surface, CJ density, tubular cristae volume, and lamellar cristae volume were counted as above. Tubular cristae volume and surface ratio were calculated as divided tubular cristae volume by sum of tubular and lamellar cristae volume to calculate tubular cristae ratio. Features were standardized by removing the mean and scaling to unit variance before use in principal component analysis (PCA) and hierarchical clustering using Ward's method with Euclidean distance. One mitochondrion with deviated size and 3 mitochondria without cristae were excluded before analysis. More details with the source code and documentation are available at: https://github.com/neurobiology-ut/PHILOW_Data_Manuscript.

### Septa structure discovery

First, mitochondria were eroded by 4 voxels. Then, when a lamellar crista structure divides mitochondria into 2 compartments, and the smaller compartment is more than 15% of the total mitochondrial volume, the structure was defined as potential septum. After a visual inspection, 4 mitochondria were selected.

## Implementation

The segmentation algorithms were trained using tensorflow 1.13.0, and the quantification algorithms were written in Python 3.7.0 using the libraries: numpy 1.17.4, opencv-python 4.0.0.21, scikit-image 0.17.2, scipy 1.3.2, and napari [61] 0.4.0. These algorithms were accelerated using the GPUs (NVIDIA TITAN RTX and NVIDIA A100) on Windows 10 with 3.6 GHz Intel Core i9-9900K processor and 32 GB memory. More details with the source code and documentation are available at: https://github.com/neurobiology-ut/PHILOW

## 3D Visualization

3D visualization of mitochondrial structure and crista structure was performed using Imaris version 9.6.0 (Bitplane).

## Supporting information

**S1 Fig. Evaluation of the mitochondrial segmentation with TAP method or HITL method.** **(A)** Segmentations of a mitochondrion with or without the TAP method. Edges of a mitochondrion were not annotated as mitochondria (magenta) by a 2D UNet++ based prediction only from the xy-plane (without TAP). With the TAP method, the same edges were successfully annotated as mitochondria by combining the predictions from all 3 axes. The green square is a fiducial marker for correlating images from different axes. **(B)** Segmentation results of mitochondria closely apposed to each other (magenta) are shown. The prediction mistakenly annotated those mitochondria as connected in the second prediction (arrowheads). At the third prediction, by adding a training data corrected from the second prediction in slice #4, misannotation in other slices were also corrected. The raw EM data are deposited in the EMPIAR (EMPIAR-11449). EMPIAR, Electron Microscopy Public Image Archive; EM, electron microscopy; TAP, three-axes prediction; 2D, two-dimensional; HITL, human-in-the-loop. (TIF)

**S2 Fig. GUI of PHILOW.** **(A)** GUI of PHILOW during annotation. The buttons for data management and the orthogonal view are shown. Low confidence areas are highlighted in green for active learning. **(B)** GUI of PHILOW showing the progress of a model training. The left graph shows the change of dice coefficient score along the number of epochs. The right graph shows the change of a loss function score along the number of epochs. By monitoring the score, users can stop the training at the moment the scores are saturated by clicking the stop button. **(C)** First, in the annotation mode, users select areas suitable for making training dataset by a button-click while observing the 3D data (i). Then, users annotate training images while looking at the orthogonal view in order to make accurate annotations (ii). The annotated images are automatically separated into original images and corresponding annotations for importing them into the model training mode (iii). The model training can be done either on a GPU machine via GUI (iv) or on Google Colaboratory (v). In the case of isotropic 3D data, our TAP method is applicable during inference, resulting in higher inference accuracy. The original images are applied to the trained model and predictions are generated (vi). The predictions are shown directly on the annotation mode for immediate proofreading (vii). The proofread slices (viii) can be added to the existing training data for the next training cycle (ix). After saturation of the prediction accuracy by iterative cycles, a final manual correction is also performed directly on PHILOW (see **Fig 3A**). The raw EM data are deposited in the EMPIAR (EMPIAR-11449). EMPIAR, Electron Microscopy Public Image Archive; EM, electron microscopy; GUI, graphical user interface; 3D, three-dimensional; GPU, graphics processing unit. (TIF)

**S3 Fig. Time measurement on PHILOW.** The same time measurement as **Fig 3C** conducted by another annotator. Source data can be found in **S14 Data**. The raw EM data are deposited in the EMPIAR (EMPIAR-11449). EMPIAR, Electron Microscopy Public Image Archive; EM, electron microscopy; HITL, human-in-the-loop; TAP, three-axes prediction; DL, deep learning.
(TIF)

**S4 Fig. Visualization of the segmentation differences between the HITL-TAP and human annotators.** Two of the 3D reconstructions by the HITL-TAP, Human #1 or Human #2 are compared. Note that the human annotators tend to segment lamellar structures more extensively than the HITL-TAP. Also note that the human annotators tend to omit tubular structures, which were recognized by the HITL-TAP (arrowheads). **(A)** Magenta (filled): HITL-TAP; Green (transparent): Human #1. **(B)** Cyan (filled): Human #2; Magenta (transparent): HITL-TAP. **(C)** Green (filled): Human #1; Cyan (transparent): Human #2. **(D)** Magenta (filled): HITL-TAP; Cyan (transparent): Human #2. **(E)** Cyan (filled): Human #2; Green (transparent): Human #1. Scale bars, 150 nm. The raw EM data are deposited in the EMPIAR (EMPIAR-11449). EMPIAR, Electron Microscopy Public Image Archive; EM, electron microscopy; 3D, three-dimensional; HITL, human-in-the-loop; TAP, three-axes prediction.
(TIF)

**S5 Fig. Shorter distance between OMM and IBM is observed next to CJs in the FIB-SEM images. (A)** The distance between the OMM (traced by blue line) and IBM (traced by red lines) was defined as the thickness of the membrane at the boundary of mitochondria. Examples of CJs in which the OMM-IBM distance was narrowed bilaterally (left), unilaterally (center), or neither (right) are shown. The graphs represent the OMM-IBM distance measured from the EM images at the bottom. Source data can be found in **S15 Data**. Scale bars, 20 nm. **(B)** The percentages of CJs in which the OMM-IBM distance was narrowed bilaterally (2), unilaterally (1), or neither (0) are shown. The raw EM data are deposited in the EMPIAR (EMPIAR-11449). EMPIAR, Electron Microscopy Public Image Archive; OMM, outer mitochondrial membranes; IBM, inner boundary membranes; CJ, crista junction; EM, electron microscopy.
(TIF)

**S6 Fig. Validation of OPA1 KD.** The amounts of OPA1 mRNA were measured by quantitative RT-PCR and normalized by Actin mRNA levels. Error bars indicate standard errors from duplicated samples of quantitative PCR. Source data can be found in **S16 Data**. $^{**}p < 0.01$, Student's $t$ test. The raw EM data are deposited in the EMPIAR (EMPIAR-11449). EMPIAR, Electron Microscopy Public Image Archive; EM, electron microscopy; OPA1, optic atrophy 1.
(TIF)

**S7 Fig. Statistical results of the structural analyses in individual control and OPA1 KD cells. (A–F)** Statistical results for each cell in control and OPA1 KD cells. Red lines show mean ± SD. Source data can be found in **S17–S22 Data**. The raw EM data are deposited in the EMPIAR (EMPIAR-11449). EMPIAR, Electron Microscopy Public Image Archive; EM, electron microscopy; OPA1, optic atrophy 1.
(TIF)

**S8 Fig. Statistical analyses of mitochondrial and crista morphology based on the parameters used in the unsupervised analysis.** Statistical analyses of the control and OPA1 KD mitochondria and cristae. **(A)** Maximum length per minimum length is shown with median (red lines). Source data can be found in **S23 Data**. $^{****}p < 0.0001$, Mann–Whitney test. **(B–I)**

Parameters of mitochondria and cristae from the control or OPA1 KD cells are indicated. Red lines show median in (B–F) and mean ± SE in (G–I). Source data can be found in **S17–S22, S24** and **S25 Data**. ***$p < 0.001$, ****$p < 0.0001$, Mann–Whitney test. The raw EM data are deposited in the EMPIAR (EMPIAR-11449). EMPIAR, Electron Microscopy Public Image Archive; EM, electron microscopy; OPA1, optic atrophy 1; PCA, principal component analysis; MCI, mitochondrial complexity index.
(TIF)

**S9 Fig. OPA1 KD cristae is more swollen than the control cristae. (A, B)** Surface area per volume of tubular (A) or lamellar (B) cristae are shown with median. Source data can be found in **S26** and **S27 Data**. ****$p < 0.0001$, Mann–Whitney test. **(C)** Representative EM images of tubular and lamellar cristae from the control and OPA1 KD cells. Note that both lamellar and tubular crista structures were thicker in the OPA1 KD mitochondria. Scale bar, 100 nm. **(D)** Representative CJs from the EM images. The red lines show CJs. **(E)** The quantification result of CJ width extracted from the EM images with $5 \times 5$ nm/px resolution. Red lines indicate median. Source data can be found in **S28 Data**. ****$p < 0.0001$, Mann–Whitney test. The raw EM data are deposited in the EMPIAR (EMPIAR-11449). EMPIAR, Electron Microscopy Public Image Archive; EM, electron microscopy; OPA1, optic atrophy 1; CJ, crista junction.
(TIF)

**S10 Fig. Long CJ from several planes.** Slot-like CJ was observed in XY plane and YZ plane. Allow heads indicate a single connected slot-like lamellar CJ. Magenta: lamellar CJ. The raw EM data are deposited in the EMPIAR (EMPIAR-11449). EMPIAR, Electron Microscopy Public Image Archive; EM, electron microscopy; CJ, crista junction.
(TIF)

**S11 Fig. The ratio of tubular cristae has no correlation with mitochondrial volume and shape in both the control and OPA1 KD. (A–C)** Statistical analyses of the tubular crista volume ratio and mitochondrial 3D structure in the control and OPA1 KD mitochondria. $R^2$ indicate the coefficient of determination. Source data can be found in **S29–S31 Data**. The raw EM data are deposited in the EMPIAR (EMPIAR-11449). EMPIAR, Electron Microscopy Public Image Archive; EM, electron microscopy; OPA1, optic atrophy 1; MCI, mitochondrial complexity index.
(TIF)

**S1 Movie. Representative 3D ultrastructure of the control mitochondria and cristae reconstructed using HITL-TAP on PHILOW.** Mitochondria and cristae were successfully segmented from the FIB-SEM stack using HITL-TAP on PHILOW. Cristae were also categorized into lamellar structure and tubular structure by HITL-TAP on PHILOW. Magenta or white: mitochondrial outer membrane, yellow: lamellar structure, cyan: tubular structure. FIB-SEM, Focused Ion Beam-Scanning Electron Microscopy; HITL, human-in-the-loop; TAP, three-axes prediction.
(MOV)

**S2 Movie. Segmentation differences between the HITL-TAP and the human annotator.** The human annotator missed to segment tubular crista structures that were well segmented by HITL-TAP method. Green: human annotator, Magenta: HITL-TAP. HITL, human-in-the-loop; TAP, three-axes prediction.
(MOV)

**S3 Movie. Representative 3D ultrastructure of the CJs.** Representative 3D reconstruction of CJs in a mitochondrion extracted by HITL-TAP method. Cyan: tubular structure, yellow:

lamellar structure, magenta: CJ. HITL, human-in-the-loop; TAP, three-axes prediction, CJ; crista junction.
(MOV)

**S4 Movie. Representative 3D ultrastructure of the OPA1 KD mitochondria and cristae reconstructed using HITL-TAP on PHILOW.** OPA1 KD mitochondria and cristae were successfully segmented from the FIB-SEM stack using HITL-TAP on PHILOW. Cristae were also categorized into lamellar structure and tubular structure by HITL-TAP on PHILOW. Magenta or white: mitochondrial outer membrane, yellow: lamellar structure, cyan: tubular structure. OPA1, optic atrophy 1; HITL, human-in-the-loop; TAP, three-axes prediction.
(MOV)

**S5 Movie. Comparison of representative 3D ultrastructure of mitochondria between the control and OPA1 KD cells.** 3D ultrastructure of 3 mitochondria each from the control cell and the OPA1 KD cells. Note that the tubular structure was significantly less in OPA1 KD mitochondria than the control ones. White: mitochondrial outer membrane, yellow: lamellar structure, cyan: tubular structure. OPA1, optic atrophy 1.
(MOV)

**S6 Movie. Representative 3D ultrastructure of mitochondria containing onion-like structures in OPA1 KD cells.** Representative 3D reconstruction of an onion-like mitochondrion. Note that the onion-like structure does not form a septum. Some parts of the inner membrane are fused to the outer membrane. White: mitochondrial outer membrane, yellow: lamellar structure, cyan: tubular structure. OPA1, optic atrophy 1.
(MOV)

**S7 Movie. Representative 3D ultrastructure of long-connected CJ.** 3D ultrastructure of a representative long CJ shown in **S10 Fig**. CJ, crista junction.
(MOV)

**S1 Data. Source data for Fig 1B.** This source file includes the F1 scores before the proofreading and the time required for the proofreading.
(XLSX)

**S2 Data. Source data for Fig 2B.** This source file includes the IoU values of the manual segmentation, the prediction using the TAP + DL baseline, or the prediction using only DL baseline. IoU, intersection over union; TAP, three-axes prediction; DL, deep learning.
(XLSX)

**S3 Data. Source data for Fig 3C.** This source file includes the time required for correcting the mitochondrial prediction results.
(XLSX)

**S4 Data. Source data for Fig 4F.** This source file includes the IoU of cristae in the manual segmentation or the prediction using the HITL-TAP method. IoU, intersection over union; HITL, human-in-the-loop; TAP, three-axes prediction.
(XLSX)

**S5 Data. Source data for Fig 6C and 6D.** This source file includes the PC1, PC2, and PC3 values of the mitochondrial properties used in PCA. PCA, principal component analysis.
(XLSX)

**S6 Data. Source data for Fig 6C and 6D.** This source file includes the PC1, PC2, and PC3 values of individual mitochondria used in PCA. PCA, principal component analysis.
(XLSX)

**S7 Data. Source data for Fig 6E.** This source file includes the data about Ward's clustering.
(XLSX)

**S8 Data. Source data for Fig 6G.** This source file includes the data about mitochondrial sub-voxel analysis.
(XLSX)

**S9 Data. Source data for Fig 7A.** This source file includes the data about lamellar angle from each lamellar.
(XLSX)

**S10 Data. Source data for Fig 7B.** This source file includes the data about lamellar angle from each mitochondrion.
(XLSX)

**S11 Data. Source data for Fig 7F.** This source file includes the data about crista junction number per inner mitochondrial membrane.
(XLSX)

**S12 Data. Source data for Fig 7G.** This source file includes the data about tubular crista junction number per tubular surface area.
(XLSX)

**S13 Data. Source data for Fig 7H.** This source file includes the data about lamellar crista junction number per lamellar surface area.
(XLSX)

**S14 Data. Source data for S3 Fig.** This source file includes the data on the times required for correcting the mitochondrial prediction results.
(XLSX)

**S15 Data. Source data for S5A Fig.** This source file includes the data about the distance between OMM and IBM. OMM, outer mitochondrial membranes; IBM, inner boundary membranes.
(XLSX)

**S16 Data. Source data for S6 Fig.** This source file includes the data about the validation of OPA1 KD. OPA1, optic atrophy 1.
(XLSX)

**S17 Data. Source data for S7A and S8B Figs.** This source file includes the data about the volume of mitochondria.
(XLSX)

**S18 Data. Source data for S7B and S8C Figs.** This source file includes the data about the mitochondria surface.
(XLSX)

**S19 Data. Source data for S7C and S8F Figs.** This source file includes the data about the crista volume.
(XLSX)

**S20 Data. Source data for S7D and S8E Figs.** This source file includes the data about the crista surface area.
(XLSX)

**S21 Data. Source data for S7E and S8H Figs.** This source file includes the data about the tubular volume ratio.
(XLSX)

**S22 Data. Source data for S7F and S8I Figs.** This source file includes the data about the tubular surface ratio.
(XLSX)

**S23 Data. Source data for S8A Fig.** This source file includes the data about the mitochondrial aspect ratio.
(XLSX)

**S24 Data. Source data for S8D Fig.** This source file includes the data about the MCI. MCI, mitochondrial complexity index.
(XLSX)

**S25 Data. Source data for S8G Fig.** This source file includes the data about the crista surface per mitochondrial volume.
(XLSX)

**S26 Data. Source data for S9A Fig.** This source file includes the data about the surface area per volume value of tubular cristae.
(XLSX)

**S27 Data. Source data for S9B Fig.** This source file includes the data about the surface area per volume value of lamellar cristae.
(XLSX)

**S28 Data. Source data for S9E Fig.** This source file includes the data about the CJ width. CJ, crista junction.
(XLSX)

**S29 Data. Source data for S11A Fig.** This source file includes the data about the mitochondrial volume and tubular volume ratio.
(XLSX)

**S30 Data. Source data for S11B Fig.** This source file includes the data about the mitochondrial aspect ratio and tubular volume ratio.
(XLSX)

**S31 Data. Source data for S11C Fig.** This source file includes the data about the MCI and tubular volume ratio. MCI, mitochondrial complexity index.
(XLSX)

## Acknowledgments

We thank Drs. Franck Polleux, Heike Blockus, Tommy L. Lewis, Jr, and Yukiko Gotoh for their critical reading of the manuscript and members of the Hirabayashi lab for constructive discussions. We would like to thank the members of LPIXEL Inc. for helpful discussions on analysis methods and advice on implementation.

## Author Contributions

**Conceptualization:** Hiroki Kawai, Yusuke Hirabayashi.

**Data curation:** Shogo Suga, Yu Nakanishi, Bruno M. Humbel.

**Formal analysis:** Shogo Suga, Koki Nakamura, Bruno M. Humbel.

**Funding acquisition:** Hiroki Kawai, Yusuke Hirabayashi.

**Investigation:** Shogo Suga, Koki Nakamura, Yusuke Hirabayashi.

**Methodology:** Shogo Suga, Koki Nakamura, Bruno M. Humbel, Hiroki Kawai.

**Software:** Hiroki Kawai.

**Supervision:** Hiroki Kawai, Yusuke Hirabayashi.

**Validation:** Yu Nakanishi.

**Writing – original draft:** Yusuke Hirabayashi.

**Writing – review & editing:** Shogo Suga, Koki Nakamura, Bruno M. Humbel, Hiroki Kawai.

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
