## [Editor Report · Decision Letter 0]

30 Sep 2022

Dear Dr Hirabayashi, 

Thank you for submitting your manuscript entitled "An interactive deep learning-based approach reveals mitochondrial cristae topologies" for consideration as a Research Article by PLOS Biology. Please also accept my apologies again for the delay in providing you with our initial decision.

Your manuscript has now been evaluated by the PLOS Biology editorial staff as well as by an academic editor with relevant expertise and I am writing to let you know that we would like to send your submission out for external peer review.

Once your full submission is complete, your paper will undergo a series of checks in preparation for peer review. After your manuscript has passed the checks it will be sent out for review. To provide the metadata for your submission, please Login to Editorial Manager (https://www.editorialmanager.com/pbiology) within two working days, i.e. by Oct 04 2022 11:59PM.

Kind regards,

Ines

--

Ines Alvarez-Garcia, PhD

Senior Editor

PLOS Biology

---

## [Decision Letter · Decision Letter 1]

9 Dec 2022

Dear Dr Hirabayashi,

Thank you for your patience while your manuscript entitled "An interactive deep learning-based approach reveals mitochondrial cristae topologies" was peer-reviewed at PLOS Biology. Thank you also for your patience as we completed our editorial process, and please accept my apologies for the delay in providing you with our decision. The manuscript has now been evaluated by the PLOS Biology editors, an Academic Editor with relevant expertise, and by three independent reviewers. 

The reviews are attached below. As you will see, the reviewers find the manuscript interesting and worth pursuing for publication, however they also ask for several additional experiments to confirm some of the cristae ultrastructure features described, the interpretation of the novel biology and roles identified for OPA1, as well as your interpretation regarding the role of lamellar versus tubular cristae. Reviewer 3 has looked exclusively at the Deep Learning aspect and requests several details that are missing on the training/test/validation data, clarifications and a comparison with existing methods.

After consulting with the Academic Editor and the rest of the team, we would like to invite you to revise the work to address the reviewers' reports. The revision should address experimentally, if at all possible, Reviewer 2's point 2 analyzing cristae structures using the new method in cell overexpressing Drp1. You should also address in detail all Reviewer 3’s comments. Qualified scientists should to be able to repeat the work if needed and you should add a clear comparison with other segmentation methods as per this reviewer final point. In addition, please recode all movies with a consistent, modern codec - H264 is recommended but at high quality; others may be appropriate but please check they are modern and appropriate.

Given the extent of revision needed, we cannot make a decision about publication until we have seen the revised manuscript and your response to the reviewers' comments. Your revised manuscript is likely to be sent for further evaluation by all or a subset of the reviewers.

**IMPORTANT - SUBMITTING YOUR REVISION**

3. Resubmission Checklist

a) *PLOS Data Policy* - IMPORTANT

b) *Published Peer Review*

Sincerely,

Ines

--

Ines Alvarez-Garcia, PhD

Senior Editor

PLOS Biology

Reviewers' comments

Rev. 1:

Powerful volumetric EM approaches such as FIB-SEM have stirred a revolution in cell biology, allowing the resolution of the organelles and even smaller ultrastructure in diverse cells. However, this method is quite demanding in terms of data acquisition and management (e.g. large file sizes) as well as the actual reconstruction of ultrastructures within the cell. The last aspect is plagued by long man hours needed to manually segment structures (i.e. outline the ultrastructure from the background) for reconstruction. Another problem with manual segmentation is biases introduced by the person modeling each of these structures. The manuscript by Suga et al., presents a very exciting and potentially revolutionary way to segment raw FIB-SEM data in a semi-automized way in conjunction with deep learning to facilitate faster and unbiased reconstruction of cristae, numerous ultrastructures that decorate the mitochondrion. This tool, which the authors have named PHILOW, is a very important advance. Remarkably, PHILOW reconstructs cristae from FIB-SEM data in a more precise and reproducible manner than human reconstructors, as also demonstrated in the manuscript.

This unbiased approach allowed the authors to reveal a significant population of tubular cristae thus far hidden from view, so far being obscured by the more prominent lamellar cristae. Furthermore, these tubular cristae are depleted upon downregulation of Opa1, a dynamin-like protein (DLP) involved in cristae remodeling and inner membrane fusion.

The paper is more or less well written presenting the PHILOW and its implementation in the analysis of Opa1 knockdowns in mouse fibroblasts. Furthermore, the microscopy images are not only informative, but visually appealing. Given the quality of the data and its presentation as well as the significant methodological advance PHILOW represents and the confirmation that tubular cristae are replete in mammalian mitochondria, I feel that the manuscript upon addressing the points below will be suitable for publication in PLoS Biology. I ask for major revision but assume the authors will have data readily available (e.g an elaboration Figure 6B) to address my point about crista junctions below. If this point is addressed, the manuscript truly represents a methodological advance that is worthy of inclusion into PLoS Biology. Thus, I do not think from the implied data that any new experiments need to be performed. Just more data about CJs should be presented among other major points.

MAJOR POINTS

1) The manuscript seems to be submitted as a Research Article. But in my opinion , it is a 'Methods and Resources' article. Given the significant methodological advance that PHILOW represent, I reiterate that upon addressing points below, the manuscript is suitable for PLoS Biology but as 'Methods and Resources' one to reflect its scope.

2) The authors claim their method allowed them to measure the "…number of CJs per mitochondria" (lines 418-9) and thus represents "…to our knowledge, this is the first report of CJ numbers and frequency." (lines 425-6 ). Aside from the scatter plot in Figure 6G and a note in the legend of Figure 6B, there is not data about observation of CJs. Given that CJs have thus far not been observed by FIB-SEM as far as I know, this would indeed be a very big advance. The authors should provide some representative images of 3D reconstructions and/or CJs revealed by segmentation on FIB-SEM sections/slices to show how CJs were revealed by PHILOW. This should also be described in Materials and Methods as for other crista ultrastructural features. Finally, the voxel size is stated to be 4.88 nm x 4.88 nm x 10 nm. CJs are about 20-25 nm in diameter on average. Thus, the CJs seem to be on the very detection limits of FIB-SEM. Thus, the authors really should explain how they observed this feature and the average CJ diameters they observed as well as their occurrence. Were CJs seen only on XY axis or also on XZ and YZ as well? If the authors can provide ample evidence for CJ detection using PHILOW, they should bring this aspect out more in front of the manuscript and not have these data buried in the Results section as it currently is, especially given the earlier here quoted claims they make in the Discussion.

3) Related to point 2 above, on lines 290-291 the authors interpret the reported widening of crista lumina as confirmation that OPA1 is involved in CJ constriction. However, since CJ width was not directly assayed in this experiment, this statement is not true. The authors should rephrase this to say their results confirm that OPA1 was a role in narrowing the width of crista lumina or even better, as per point 2, report the measured widths of CJs in this experiment. Once again, I think that the potential capacity of PHILOW to assay CJ properties in a high-throughput way is very exciting.

4) The final section of the Results (starting on Line 345), the authors compare their results on OPA1 RNAi-silencing in mouse fibroblasts with results reported in other articles in a somewhat misleading way. First of all, discussed references 33, 41 and 42 studied the inner membrane DLP MGM1 in baker's yeast, not OPA1 per se. While these two proteins likely perform the same or overlapping functions in IM fusion and crista maintenance, they may not be true homologs. This issue of course is outside the scope of the paper, but the authors should at least say which organism these data were derived from and mention the yeast name of the protein. Second, the authors write "…OPA1 KD also produces mitochondria with an onion-like structure [33]". The cited reference did not use OPA1 KD as written but a temperature-sensitive mutant of MGM1. AS far as I know when the yeast is put in the non-permissive temperature of 37°C, the mitochondrial fusion is inhibited, presumably due to point mutations in the GTPase domain. This is not a knockdown not a knockdown per se. Third, prior to this the term 'Opa1 KD' was used in the context of septum formation. Again, reference 33 uses the temperature-sensitive MGM1 mutant, reference 41 does not mention septa at all, and reference 42 uses a mgm1 knockouts in strains where other DLPs are also deleted. Thus, references 41 and 42 are not appropriate here and in fact the authors state the fzo deletion leads to more septa formation than mgm1 deletion in reference 42. Finally, the authors discuss the phenotype of a site mutation in the C. elegans EAT-1 protein. They name this properly but fail to mention this is site mutation in the GTPase domain and not a RNAi-silencing. Thus, the authors should correctly discuss their OPA1 KDs phenotypes in the context of these various genotypes/conditions/organisms.

5) The way the data in the figures in written about in the results is done so in a very confusing way. For example Figure 3D is mentioned before Figure 3C on page 6. Figure 5 is even more confusingly presented, with 5F mentioned first and then 5A. Furthermore, the following main figures are never referenced in the manuscript: Fig 2E, 2F , 6A and 6B. The authors should order their figures in the order they are mentioned in the manuscript and mention ALL figure elements. Please also treat the supplemental Figures the same way.

6) Data in 6A can go into supplement as it only verifies OPA1 RNAi and thus is a control. Figure 6B notes that the CJ area is less but not indicated at all. This refers to Point 2 above. The CJs should be highlighted and the sections with CJs of these images should be shown, as also mentioned in point 2.

7) Tables 1-4 are not contain any descriptive titles or small legend.

8) I could not open Videos 1,2 and 4 as there were issues with not having the proper codec. I did not have such issues with the other videos. Please be sure Videos 1,2 and 4 use the same codec as Videos 3 and 5.

9) The paper seems to focus mostly on the application of PHILOW on mitochondria with tubular and lamellar cristae. However, FIB-SEM has been used to observe the unique cristae of algae (Uwizeye et al., 2021 PNAS 118 (27): e2025252118) and trypanosomes (Bily et al., 2020 J Euk Micro 68: e12846). The authors should at least mention whether they think PHILOW will work on other crista shapes, especially hard to discern ones sometimes found in such organisms.

10) Lines 258-61. Authors assay 5 cells from control and OPA1 KD, observing 135 and 324 mitochondria respectively. I would appreciate if the authors had a plot indicating average mitochondria per cell with standard deviation.

11) What was the rationale of the authors reporting surface area/volume of mitochondria (Lines 264-7, Table 1)?

12) Pertaining to point 2 above. According to the methods, the the acquired FIB-SEM data had a final voxel size of 4.88 nm x 4.88 nm x 10 nm (line 686). Why is the z axis different? Is this really isotropic?

Minor points:

1) Abbreviations in Figures should be defined in their respective legends (e.g. IoU).

2) Lines 2675-8: "…a gene responsible for optic atrophy." Citation missing.

3) Line 265, should read "…mean mitochondrial surface area…"

4) Throughout text, if used as adjective, use singular form 'crista' (e.g. lines 383, 395)

5) In Methods, delete 'for' in e.g. "…for 3 times…" (line 657). Also, please either use numerals or words for numbers but do not mix these 2 forms.

6) Line 720, should read "…with the following parameters:"

Rev. 2:

The manuscript by Suga, Nakamura et al report a novel image analysis method for FIB-SEM micrographs that decrease processing time, named PHILOW. This new method allowed for a higher throughput 3D reconstruction of mitochondrial cristae, segmenting in a manner that could even supersede the capacity of the human eye. Using this method, they conclude that an underappreciated defect caused by OPA1 knock down is a change in cristae orientation, as well as in the ratio of lamellar and tubular cristae. In addition, authors claim that this is the first time reporting total cristae junction numbers and frequency. The method seems robust and provides a significant advancement in terms of speed and segmentation. In addition, authors made it publicly available, which is great. However, the main concerns related to the interpretation of the novel biology and roles identified for OPA1, as well as the interpretation of the authors about the role of lamellar versus tubular cristae.

1) Most of the data shown should be binned by individual mitochondria, as there is a possibility that differences in mitochondrial size are driving most of the phenotypes. Authors should show whether there is any correlation between number and frequency of cristae junctions, lamellar and tubular mitochondria with mitochondrial aspect ratio, form factor or size. Among the heterogenous structures of control mitochondria (Figure 5B), one could see that there is a correlation between of how the cristae are organized depending on the shape of the individual. Therefore, is OPA1 changing these novel parameters because it is inducing fragmentation or because it is truly reflecting a specific action of OPA1 in the cristae.

2) Authors should analyze cristae structures using their new method in cells with shorter mitochondria without inducing OPA1 cleavage (i.e. Drp1 overexpression). Such an experiment is required to prove the authors conclusions of OPA1 being responsible for controlling these new aspects of mitochondria cristae.

3) Authors claimed that they could count for the first time the number of CJ. Could they quantify mitochondrial contact sites as well (area where the distance between the outer and inner membrane is reduced)?

Minor comments: the paper include inaccurate statements such as:

1) "Since diffusion of ions and metabolites through the narrow compartment of cristae represents a significant rate-limiting step of biochemical reactions in mitochondria" This is not true, diffusion of ions is not the rate limiting step. In the case of OXPHOS, is ADP availability in the matrix.

2) "Since the orientation of cristae directly affects the distance metabolites need to travel from the cytosol to biochemical reaction sites on IMM, this finding raises the possibility that cristae orientation is regulated to meet metabolic needs of ever-changing cellular conditions." There are many inaccuracies: a)orientation does not directly affect distance, b) it could be cristae length and distance to the outer membrane and c) how orientation of the cristae is the best way to regulate ever changing conditions. I would strongly encourage the authors to remove this sentence.

Rev. 3:

The paper proposed a human-in-the-loop object segmentation approach to study mitochondrial cristae topologies. My comments are concerned with the deep learning part of the manuscript. I can't judge the biological significance of the finding.

The proposed method has three key elements:

1. A 2d UNet++ for predicting segmentations

2. Human-in-the-loop for prediction correction and adding training

3. A 3D integration step to improve robustness

The authors demonstrated impressive F1-score performance.

Comments

1. There are very few details about the deep learning methods. More details on the training/test/validation data are needed to better understand the performance.

2. Does every human-in-the-loop cycle need to retrain or fine-tune the model?

3. What do you mean by a probability of 127 or [1, 126]?

4. It is not clear if any of the F1-score is based on test data.

5. How robust is the method against different human proofreaders? The authors showed two, and some nontrivial differences can already be observed.

6. Segmenting objects from EM images has been a task attempted by many deep learning approaches. The authors cited a few applied to connectomes. No comparison with those existing methods are showed.

---

## [Decision Letter · Decision Letter 2]

3 May 2023

Dear Dr Hirabayashi,

Thank you for your patience while we considered your revised manuscript entitled "An interactive deep learning-based approach reveals mitochondrial cristae topologies" for consideration as a Research Article at PLOS Biology. Your revised study has now been evaluated by the PLOS Biology editors, the Academic Editor and the three original reviewers. 

The reviews are attached below. The reviewers appreciate all the work you have done in the revision and Reviewer 3 is now satisfied. Nevertheless, both Reviewers 1 and 2 raise several remaining points that would need to be addressed. Reviewer 1 would like you to clarify several points, whereas Reviewer 2 asks for further analyses to measure the aspect ratio and form factors, rather than the volume of individual mitochondria. In addition, this reviewer would like you to integrate in the paper some data that has been only added to the rebuttal.

After discussing the reviews with the Academic Editor, we think you should address all the remaining points. We do not think that the new analyses requested would take you very long, but if you don’t perform these, please justify in detail why they are not done. We will assess your revised manuscript and your response to the reviewers' comments with our Academic Editor aiming to avoid further rounds of peer-review, although might need to consult with the reviewers, depending on the nature of the revisions.

**IMPORTANT - SUBMITTING YOUR REVISION**

*Resubmission Checklist*

Sincerely,

Ines

--

Ines Alvarez-Garcia, PhD

Senior Editor

PLOS Biology

Reviewers' comments

Rev. 1:

The revision by Suga et al., addresses the issues that I raised, but unfortunately brings up another issue in the process of addressing the crista junctions (CJs). I should say that I do appreciate particularly how the authors addressed points about CJ analysis, addressed in Figures 5 and S5 of the revision.

Although I think the revision is much improved after their revision, the issue with the interpretation of the CJs the authors rendered on lamellar cristae must be addressed before acceptance of the paper to PLoS Biology. This and some other major points are listed below, followed by some minor language corrections.

I reiterate that this manuscript reports an important tool for rapid and robust analysis of mitochondrial ultrastructure, revealing that mammalian mitochondria bear tubular and lamellar cristae, whose balance is affected by OPA1. But the potential limitation of FIB-SEM visualization of CJs (affecting PHILLOW processing) I point out below must be addressed.

Major points:

1) The very long, slit-like CJs reported in the aforementioned figures and described in the chapter starting on line 261 contradict observations of CJs rendered from electron tomograms in MEFs and other metazoans. Here the authors observe very long CJs at the base of lamellar cristae. In contrast, in all electron-tomographic observations I know of, the base of lamellar cristae are attached to the inner boundary membrane (IBM) by multiple tubular or in some cases oval CJs that are often adjacent to each other.

The authors should at least acknowledge that these PHILLOW renderings of CJs in lamellar do not agree with tomogram-renderings of the CJs (multiple papers listed in the cited Panek review). Perhaps this is an artifact of the electron beam voltage (stated on line 740), which observes structures ~20 nm deep into the sample, which is more than the 10 nm thick milling sections (line 739). This may result in the appearance of contact along the lamellar crista border with the IBM, when in reality there is a small gap between parallel crista and IBM membranes in between a string of adjacent tubular/oval CJs along this border. For example, the second crista from left shown in Figure 5C middle panel agrees with tomogram renderings of CJs, but the two most right ones do not.

It is very hard to evaluate the rendered CJ models without any of the raw images of the FIB-SEM section that contains this/these CJs. The authors can also show this data to support their long slit-CJs or demonstrate any underlying limitations for rendering adjacent CJs that can be addressed by future optimization.

This also raises question for me about how CJ numbers were calculated in Fig 7E. Were these long CJs potentially arising from the aforementioned artifact considered a single crista? Since tubular cristae are convincingly shown to be reduced in OPA1 knockdowns (KDs), and these seem to have conventional tubular CJs (Fig 5), could the increase in long CJs of lamellar cristae be true reason for 1) less CJs and/or 2) apparently wider CJs?

If not done so already, I would suggest that CJ measurements be made exclusively on a specific crista type. The tubular cristae may be the best target given the problems with the long CJs of lamellar cristae.

2) This and the next point are less major than the previous point. The authors in several points in the manuscript state that Mic10 and Mic60 ablation results in onion-like cristae (lines 412 and 508). The phenotypes of these two genes are stacked detached cristae often longer in length. Perhaps the onion-like cristae are seen occasionally in these mutants, but this phenotype typifies yeast cells in which ATP synthase dimerization is impaired (e.g. Paumard et al, 2002 EMBO J 21:221-30).

As per this comment, I do not follow the logic presented by the authors from line 413, somehow correlating CJ density and the appearance of onion-like structures in OPA1-silenced cells. I agree with the authors statement that this represents a deformed IM, most likely due to a IM fusion defect. However, I still do not see how this is connected with CJ formation, and would report these data in a different context.

3) On lines 427-8, the authors write how crista contribute to "biochemical reactions is speculative". True, there is much debate about whether cristae increase the surface area of the mitochondria to give eukaryotes more ATP per gene, but I also think work such as the cited Wolf article (2019, EMBO J) nicely demonstrate the notion these are bioenergetic sub-compartments within the organelle. So I do not think the function of cristae as bioenergetic subcompartments is speculative, but well established by this and other studies showing enrichments of respiratory chain complexes in crista membranes.

Minor Language Correction:

Line 126: "less proofreading time"

Line 228: "of algae and trypanosomes"

Line 229: "laborious,"

Line 254: rather use same 3 decimal recall values as in Fig 4C.

Line 262-3: CJ is rather "a narrow ~20 nm attachment point of a crista to the inner boundary membrane (IBM)" They have been also described as 'narrow necks' or something like this in the literature.

Line 321: rather "cellular respiration"

Line 394: "of Mgm1p"

Line 403: "suggested that"

Rev. 2:

The authors partially addressed the major concerns, but not successfully. Rather than the volume of individual mitochondria , the aspect ratio and form factors needed to be measured, as aspect ratio and form factor are parameters that quantify shape. Accordingly, the tubular shape of cristae can still be dependent on the change of shape of individual mitochondria induced by OPA1 KD, rather than by OPA1 directly determining tubular shape. With the Drp1 overexpression experiment being missing, this remains an open question. These analysess must be presented in the manuscript (not only in the rebuttal) but using aspect ratio as the parameter to bin and perform correlations and PCA analyses. The minor concers were not addressed: bottleneck can be considered as a synonim to rate limiting. Thus the authors did not perform the change requested. The same issue occurred with the second sentence about orientation.

Rev. 3:

I thank the authors for throughly addressing all of my points. I'm happy for the manuscript to be published in this revised version.

---

## [Editor Report · Decision Letter 3]

22 Jun 2023

Dear Dr Hirabayashi,

Thank you for your patience while we considered your revised manuscript entitled "An interactive deep learning-based approach reveals mitochondrial cristae topologies" for publication as a Research Article at PLOS Biology. This revised version of your manuscript has been evaluated by the PLOS Biology editors and by the Academic Editor.

Based on our Academic Editor's assessment of your revision, we are likely to accept this manuscript for publication, provided you satisfactorily address the data and other policy-related requests stated below.

We expect to receive your revised manuscript within two weeks. 

*Published Peer Review History*

*Press*

Sincerely,

Ines

--

Ines Alvarez-Garcia, PhD

Senior Editor

PLOS Biology

Fig. 1B; Fig. 2B; Fig. 3C; Fig. 4F; Fig. 6C, D, E, G; Fig. 7A, B, F-H; Fig. S3; Fig. S5A; Fig. S6; Fig. S7A-F; Fig. S8A-I; Fig. S9A, B, E and Fig. S11A-C

If any of the data shown in the graphs is deposited in EMPIAR, please indicate so in the figure legend including the URL.

---

## [Editor Report · Decision Letter 4]

12 Jul 2023

Dear Dr Hirabayashi,

Thank you for the submission of your revised Research Article entitled "An interactive deep learning-based approach reveals mitochondrial cristae topologies" for publication in PLOS Biology. On behalf of my colleagues and the Academic Editor, Robert Insall, I am delighted to let you know that we can in principle accept your manuscript for publication, provided you address any remaining formatting and reporting issues. These will be detailed in an email you should receive within 2-3 business days from our colleagues in the journal operations team; no action is required from you until then. Please note that we will not be able to formally accept your manuscript and schedule it for publication until you have completed any requested changes.

PRESS

Sincerely, 

Ines

--

Ines Alvarez-Garcia, PhD

Senior Editor

PLOS Biology
